# Connecting Federated ADMM to Bayes

**Siddharth Swaroop**
Harvard University, US
`siddharth@seas.harvard.edu`

**Mohammad Emtiyaz Khan**
RIKEN Center for AI Project, Japan

**Finale Doshi-Velez**
Harvard University, US

## ABSTRACT

We provide new connections between two distinct federated learning approaches based on (i) ADMM and (ii) Variational Bayes (VB), and propose new variants by combining their complementary strengths. Specifically, we show that the dual variables in ADMM naturally emerge through the "site" parameters used in VB with isotropic Gaussian covariances. Using this, we derive two versions of ADMM from VB that use flexible covariances and functional regularisation, respectively. Through numerical experiments, we validate the improvements obtained in performance. The work shows connection between two fields that are believed to be fundamentally different and combines them to improve federated learning.

## 1 INTRODUCTION

The goal of federated learning is to train a *global* model in the central server by using the data distributed over many *local* clients (McMahan et al., 2016). Such distributed learning improves privacy, security, and robustness, but is challenging due to frequent communication needed to synchronise training among nodes. This is especially true when the data quality differs drastically from client to client and needs to be appropriately weighted. Designing new methods to deal with such challenges is an active area of research in federated learning.

We focus on two distinct federated-learning approaches based on the Alternating Direction Method of Multipliers (ADMM) and Variational Bayes (VB), respectively. The ADMM approach synchronises the global and local models by using constrained optimisation and updates both primal and dual variables simultaneously. This includes methods like FedPD (Zhang et al., 2021), FedADMM (Gong et al., 2022; Wang et al., 2022; Zhou and Li, 2023) and FedDyn (Acar et al., 2021). The VB approach, on the other hand, uses local posterior distributions as messages and multiplies them to compute an accurate estimate of the global posterior (Ashman et al., 2022). This follows the more general Bayesian framework which has a long history in distributed and decentralised computations (Mutambara, 1998; Tresp, 2000; Durrant-Whyte, 2001). Despite decades of work, the two fields remain disconnected and are thought to be fundamentally different: there is no notion of duality in VB, and posterior distributions are entirely absent in ADMM. Little has been done to connect the two at a fundamental level and our goal here is to address this gap.

In this paper, we provide new connections between ADMM and VB-based approaches for federated learning. Our main result shows that the dual-variables used in ADMM naturally emerge through the "site" parameters in VB. We show this for a specific case where isotropic-covariance Gaussian approximations are used. For this case, we get a close line-by-line correspondence between the ADMM and VB updates. The expectations of the local likelihood approximations, also known as the sites, yield the dual terms used in the local updates of ADMM.

We use this connection to derive new federated learning algorithms. First, we extend the isotropic case to learn full-covariances through the federated VB approach; see Sec. 3.3. This leads to an ADMM-like approach with an additional dual variable to learn covariances, which are then used as preconditioners for the global update. Second, building upon functional regularisation approaches from Bayesian literature, we propose to use similar extensions to handle unlabelled data, providing a relationship to federated distillation methods (Seo et al., 2020; Li and Fedmd, 2019; Lin et al., 2020a;

Seo et al., 2020; Wu et al., 2021), as well as work in function-space continual learning (Buzzega et al., 2020; Kirkpatrick et al., 2017; Pan et al., 2020; Titsias et al., 2020); see Sec. 3.4. Our empirical results in Sec. 4 show that the new algorithms lead to improved performance (improved convergence) across a range of different models, datasets, numbers of clients, and dataset heterogeneity levels.

## 2 Federated Learning with ADMM and Bayes

We start by introducing federated ADMM and Bayesian federated learning approaches, and then, in Secs. 3 and 4, we will provide new connections along with new algorithms and empirical results. The goal of federated learning is to train a *global* model (parameters denoted by $\boldsymbol{w}_g$) by using the data distributed across $K$ clients. Ideally, we want to recover the solution that minimises the loss over all the aggregated data, for instance, we may aim for

$$\boldsymbol{w}_g^* = \arg\min_{\boldsymbol{w}} \sum_{k=1}^K \ell_k(\boldsymbol{w}) + \delta \mathcal{R}(\boldsymbol{w}), \tag{1}$$

where $\ell_k$ is the loss defined at the $k$'th client using its own local data $\mathcal{D}_k$ and $\mathcal{R}$ is a regulariser with $\delta \geq 0$ as a scalar (set to 0 when no explicit regularisation is used). However, because the data is distributed, the server needs to query the clients to get access to the losses $\ell_k$. Then the goal is to estimate $\boldsymbol{w}_g^*$ while also minimising the communication cost. Typically, instead of communicating $\ell_k$ or its gradients, local clients train their own models (denoted by $\boldsymbol{w}_k$ for the $k$'th client) and communicate them (or changes to them) to the server. The server then updates $\boldsymbol{w}_g$ and sends it back to the clients. This process is iterated with the aim of eventually recovering $\boldsymbol{w}_g^*$.

The Alternating Direction Method of Multipliers (ADMM) is a popular framework for distributed and federated learning, where the local and global optimisation are performed with constraints that they must converge to the same model. That is, we use constraints $\boldsymbol{w}_k = \boldsymbol{w}_g$ enforced through the dual variables, denoted by $\boldsymbol{v}_k$. The triple $(\boldsymbol{w}_k, \boldsymbol{v}_k, \boldsymbol{w}_g)$ are updated as follows,

$$\begin{aligned} \text{Client updates:} \quad \boldsymbol{w}_k &\leftarrow \underset{\boldsymbol{w}}{\arg\min} \ \bar{\ell}_k(\boldsymbol{w}) + \boldsymbol{v}_k^\top \boldsymbol{w} + \tfrac{1}{2}\alpha\|\boldsymbol{w} - \boldsymbol{w}_g\|^2, \\ \boldsymbol{v}_k &\leftarrow \boldsymbol{v}_k + \alpha(\boldsymbol{w}_k - \boldsymbol{w}_g), \quad \text{for all } k, \\ \text{Server update:} \quad \boldsymbol{w}_g &\leftarrow \frac{1}{K} \sum_{k=1}^K \left[ \boldsymbol{w}_k + \frac{1}{\alpha} \boldsymbol{v}_k \right]. \end{aligned} \tag{2}$$

Here, $\alpha > 0$ is a scalar and we denote $\bar{\ell}_k = \ell_k/N_k$ where $N_k$ is the size of the data at the $k$'th client. Upon convergence, we get $\boldsymbol{w}_g^* = \boldsymbol{w}_k^*$ and $\boldsymbol{v}_k^* = -\nabla \bar{\ell}_k(\boldsymbol{w}_g^*)$ for all $k$.

Such updates are instances of primal-dual algorithms and many variants have been proposed for federated learning (Zhang et al., 2021; Gong et al., 2022; Wang et al., 2022; Zhou and Li, 2023). For example, FedDyn (Acar et al., 2021) is perhaps the best-performing variant which uses a client-specific $\alpha_k \propto \alpha/N_k$. It also adds an additional hyperparameter through a local weight-decay regulariser added to each local client's loss. There are also simpler versions (not based on ADMM), for instance, FedProx (Li et al., 2020) where $\boldsymbol{v}_k$ is not used (or equivalently fixed to 0 in Eq. 2), and even simpler Federated Averaging or FedAvg (McMahan et al., 2016), where the proximal constraint $\boldsymbol{w} - \boldsymbol{w}_g$ is also removed. Such methods do not ensure that $\boldsymbol{w}_k^* = \boldsymbol{w}_g^*$ at convergence. The use of dual-variables is a unique feature of ADMM to synchronise the server and clients.

The Bayesian framework for federated learning takes a very different approach to ADMM. Instead of a point estimate $\boldsymbol{w}_g^*$, the goal for Bayes is to target a global posterior $p_g(\boldsymbol{w}) = p(\boldsymbol{w}|\mathcal{D}_1, \mathcal{D}_2, \ldots, \mathcal{D}_K)$. Instead of loss functions for each $\mathcal{D}_k$, we use likelihoods denoted by $p(\mathcal{D}_k|\boldsymbol{w})$. The solution $\boldsymbol{w}_g^*$ in Eq. 1 can be seen as the mode of the posterior $p_g$ whenever there exists likelihood such that $\log p(\mathcal{D}_k|\boldsymbol{w}) = -\ell_k(\boldsymbol{w})$ and a prior $\log p_0(\boldsymbol{w}) = -\delta\mathcal{R}(\boldsymbol{w})$, where both equations need to be true only up to a constant. Therefore, targeting the posterior $p_g(\boldsymbol{w})$ also recovers $\boldsymbol{w}_g^*$. Often, we compute local posteriors $p_k(\boldsymbol{w}) = p(\boldsymbol{w}|\mathcal{D}_k)$ at clients and combine them at the servers, for instance, Bayesian committee machines (Tresp, 2000) use the following update,

$$p_g \propto p_0 \prod_{k=1}^K t_k. \tag{3}$$

where $t_k(\boldsymbol{w}) = p_k(\boldsymbol{w})/p_0(\boldsymbol{w})$ is essentially equivalent to the likelihood $p(\mathcal{D}_k|\boldsymbol{w})$ over the data $\mathcal{D}_k$ at the client $k$. However, the data is never communicated and only the local posterior $p_k$ is passed. Unlike ADMM, the update is not iterative because the posteriors $p_k$ are assumed to be exact, giving rise to a one-step closed-form solution for $p_g$. Earlier work on distributed and decentralised Bayes focused on such cases (Mutambara, 1998; Durrant-Whyte, 2001; Scott et al., 2022). This is perhaps the main reason why Bayesian updates can seem disconnected from the ADMM-style updates.

We will derive a connection by using *approximate* Bayesian approaches, which are required to scale Bayes to large complex models. Specifically, we will use Variational Bayes (VB), although there also exist work on Monte Carlo methods (Al-Shedivat et al., 2021; Deng et al., 2024; Liang et al., 2024). VB finds an approximate posterior, denoted by $q_g(\boldsymbol{w})$, by minimising the Kullback-Leibler (KL) divergence $\mathbb{D}_{KL}[q_g \,\|\, p_g]$. For exponential-family distributions, the global $q_g(\boldsymbol{w})$ has a similar form to Eq. 3 but $t_k(\boldsymbol{w})$ are replaced by likelihood approximations $\hat{t}_k(\boldsymbol{w})$, also known as "sites". For instance, when the prior $p_0(\boldsymbol{w})$ has the same exponential-form as $q_g(\boldsymbol{w})$, the optimal $q_g^*(\boldsymbol{w})$ is

$$q_g^* \propto p_0 \prod_{k=1}^{K} \hat{t}_k^*. \tag{4}$$

The sites $\hat{t}_k^*(\boldsymbol{w})$ in VB are parameterised through natural gradients as reported by Khan and Lin (2017, Eq. 11); see an exact expression in Khan and Nielsen (2018, Eq. 18) or Khan and Rue (2023, Sec. 5.4). In this paper, we will show that a distributed estimation of $\hat{t}_k^*(\boldsymbol{w})$ leads to a natural emergence of dual variables and an ADMM-like algorithm. Specifically, we will rely on the Partitioned Variational Inference (PVI) procedure of Ashman et al. (2022) and show that, for a Gaussian $q_g(\boldsymbol{w})$, we get a line-by-line correspondence to ADMM. We note that the other prominent approach called Expectation Propagation (EP) (Minka, 2001; Guo et al., 2023) also obtains site parameters by optimising the *forward* KL, $\mathbb{D}_{KL}[p_g \,\|\, q_g]$, but there is no known connection to the dual variables.

## 3 CONNECTING VARIATIONAL BAYES TO ADMM

We precisely connect Variational Bayes, specifically Partitioned Variational Inference (PVI) (Ashman et al., 2022), to ADMM-style methods for federated learning. We start with PVI, noting it has similar components to ADMM. We then make approximations to derive a method (FedLap) that has close line-by-line correspondence to ADMM. We use this connection to derive new variants of ADMM to improve it, by (i) using full covariance information, and (ii) including function-space information.

### 3.1 CONNECTING PARTITIONED VARIATIONAL INFERENCE (PVI) TO ADMM

Partitioned Variational Inference (PVI) (Ashman et al., 2022; Bui et al., 2018) aims to find the best approximation $q_g \in \mathcal{Q}$ where $\mathcal{Q}$ is a set of candidate distributions (for example, a set of Gaussian distributions). The goal is to get the VB solution in Eq. 4 but by using an iterative message passing algorithm where local approximations $q_k(\boldsymbol{w})$ send the sites $\hat{t}_k(\boldsymbol{w})$ as messages to improve $q_g(\boldsymbol{w})$. The method updates the following triple $(q_k, \hat{t}_k, q_g)$ as shown below,

$$\text{Client updates:} \quad q_k \leftarrow \arg\max_{q \in \mathcal{Q}} \ \mathbb{E}_q\left[\log \frac{p(\mathcal{D}_k|\boldsymbol{w})}{\hat{t}_k(\boldsymbol{w})}\right] - \mathbb{D}_{KL}[q \,\|\, q_g],$$

$$\hat{t}_k \leftarrow \hat{t}_k\left(\frac{q_k}{q_g}\right), \tag{5}$$

$$\text{Server update:} \quad q_g \propto p_0 \prod_{k=1}^{K} \hat{t}_k.$$

The update of $q_g(\boldsymbol{w})$ is exactly the same as Eq. 3 but uses $\hat{t}_k(\boldsymbol{w})$ which in turn is obtained by using the ratio $q_k(\boldsymbol{w})/q_g(\boldsymbol{w})$ in the second line. The $\hat{t}_k(\boldsymbol{w})$ essentially uses the discrepancy between the global and local distributions. This is then used in the first line to modify the local $q_k(\boldsymbol{w})$. Similarly to ADMM, at convergence we have $q_g^*(\boldsymbol{w}) = q_k^*(\boldsymbol{w})$. The site parameters are related to natural gradients (Khan and Lin, 2017; Khan and Nielsen, 2018), therefore we may expect them to be related to dual variables $\boldsymbol{v}_k^*$ in ADMM which estimate gradients. In what follows, we will derive this precisely.

The PVI update has a line-by-line correspondence to ADMM. We replace $\log p(\mathcal{D}_k|\boldsymbol{w})$ by $-\ell_k(\boldsymbol{w})$ and turn the max in the first line into a min, and write the second and third lines in log-space,

$$\text{Client updates:} \quad q_k \leftarrow \arg\min_{q \in \mathcal{Q}} \ \mathbb{E}_q[\ell_k(\boldsymbol{w})] + \mathbb{E}_q[\log \hat{t}_k] + \mathbb{D}_{KL}[q \,\|\, q_g]$$

$$\log \hat{t}_k \leftarrow \log \hat{t}_k + \rho \left(\log q_k - \log q_g\right) \tag{6}$$

$$\text{Server update:} \quad \log q_g \leftarrow \log p_0 + \sum_{k=1}^{K} \log \hat{t}_k + \text{const},$$

where we have added a damping factor $0 < \rho \le 1$ as it can slow down the rate of change of $\hat{t}_k(\boldsymbol{w})$, which can be important especially in heterogenous settings (Ashman et al., 2022), although the theory suggests that $\rho = 1$ should be ideal. The three updates have a similar form to the ADMM updates in Eq. 2. The first line uses an additional expectation over $q(\boldsymbol{w})$ with the Euclidean distance replaced by the KL divergence. The role of $\hat{t}_k(\boldsymbol{w})$ appears to be similar to that of the dual term $\boldsymbol{v}_k^\top \boldsymbol{w}$ in ADMM. The update of $\hat{t}_k(\boldsymbol{w})$ in the second line is also very similar, while the update of $q_g(\boldsymbol{w})$ has some major differences. Next, we will choose the distributions in PVI accordingly to get even closer to the ADMM update.

## 3.2 FEDLAP: A LAPLACE VERSION OF PVI

We now choose a specific form of the distributions in PVI to make it even closer to the ADMM update, deriving a new method which we call FedLap. In Secs. 3.3 and 3.4 we will use this connection to derive new, improved, variants of ADMM. Here, we set the family $\mathcal{Q}$ to be the set of isotropic Gaussian distributions $\mathcal{N}(\boldsymbol{w}; \boldsymbol{m}, \mathbf{I}/\delta)$ where the mean $\boldsymbol{m}$ needs to be estimated while the covariance is fixed to $\mathbf{I}/\delta$ where $\delta > 0$ is a scalar. We also set the prior $p_0(\boldsymbol{w})$ to a zero mean Gaussian with the same precision as $q(\boldsymbol{w})$. These choices are shown below,

$$q_g(\boldsymbol{w}) \propto \mathcal{N}(\boldsymbol{w}; \boldsymbol{w}_g, \mathbf{I}/\delta), \qquad q_k(\boldsymbol{w}) \propto \mathcal{N}(\boldsymbol{w}; \boldsymbol{w}_k, \mathbf{I}/\delta), \qquad p_0(\boldsymbol{w}) \propto \mathcal{N}(\boldsymbol{w}; 0, \mathbf{I}/\delta). \tag{7}$$

These choices imply that $\hat{t}_k(\boldsymbol{w})$ takes a Gaussian form where only the linear term needs to be estimated, that is, we need to find $\boldsymbol{v}_k$ such that,

$$\hat{t}_k(\boldsymbol{w}) \propto e^{\delta \boldsymbol{v}_k^\top \boldsymbol{w}}. \tag{8}$$

This form is ensured due to the form of the optimal solution in Eq. 4 (and can be shown more formally by using natural gradients). Roughly speaking, this is because both $q_g(\boldsymbol{w})$ and $p_0(\boldsymbol{w})$ have the exact same Gaussian form, therefore $\delta$ also appears in the expression of $\hat{t}_k(\boldsymbol{w})$ as well.

Plugging these in Eq. 6 and making a delta approximation (Khan and Rue, 2023, App. C.1), we get FedLap, a Laplace variant of the PVI (we first provide the update, and then its derivation),

$$\text{Client updates:} \quad \boldsymbol{w}_k \leftarrow \arg\min_{\boldsymbol{w}} \ \ell_k(\boldsymbol{w}) + \delta \boldsymbol{v}_k^\top \boldsymbol{w} + \tfrac{1}{2}\delta\|\boldsymbol{w} - \boldsymbol{w}_g\|^2$$

$$\boldsymbol{v}_k \leftarrow \boldsymbol{v}_k + \rho(\boldsymbol{w}_k - \boldsymbol{w}_g) \tag{9}$$

$$\text{Server update:} \quad \boldsymbol{w}_g \leftarrow \sum_{k=1}^{K} \boldsymbol{v}_k.$$

The first line is obtained by making the delta approximation, that is, $\mathbb{E}_{q \sim \mathcal{N}(\boldsymbol{w}; \boldsymbol{m}, \mathbf{I}/\delta)}[g(\boldsymbol{w})] \approx g(\boldsymbol{m})$,

$$\mathbb{E}_q[\ell_k(\boldsymbol{w})] + \mathbb{E}_q[\log \hat{t}_k(\boldsymbol{w})] + \mathbb{E}_q\left[\log \frac{q(\boldsymbol{w})}{q_g(\boldsymbol{w})}\right] \quad \approx \quad \ell_k(\boldsymbol{m}) + \delta \boldsymbol{v}_k^\top \boldsymbol{m} + \tfrac{1}{2}\delta\|\boldsymbol{m} - \boldsymbol{w}_g\|^2 + \text{const}.$$

Then, rewriting $\boldsymbol{m}$ as $\boldsymbol{w}$, we get the first line. The second line follows due to the Gaussian form,

$$\delta \boldsymbol{v}_k^\top \boldsymbol{w} \leftarrow \delta \boldsymbol{v}_k^\top \boldsymbol{w} + \rho \left(\delta \boldsymbol{w}_k^\top \boldsymbol{w} - \delta \boldsymbol{w}_g^\top \boldsymbol{w}\right) \quad \implies \quad \boldsymbol{v}_k \leftarrow \boldsymbol{v}_k + \rho\left(\boldsymbol{w}_k - \boldsymbol{w}_g\right).$$

The last line follows in the same fashion where we update the mean $\boldsymbol{w}_g$ of $q_g(\boldsymbol{w})$.

The derivation shows clearly that the $\hat{t}_k(\boldsymbol{w})$ terms used in the KL minimisation of PVI gives rise to the term $\boldsymbol{v}_k^\top \boldsymbol{w}$ in the ADMM update in Eq. 2. Similarly to ADMM and PVI, at convergence we have $\boldsymbol{w}_g^* = \boldsymbol{w}_k^*$. We can make the FedLap update look even more similar to the ADMM update by a change of variable, absorbing $\delta$ into $\boldsymbol{v}_k$ (see Eq. 18 in App. A). We also note three subtle differences.

1. First, instead of using $\alpha$ to update both $\boldsymbol{w}_k$ and $\boldsymbol{v}_k$, here we use two separate parameters. The Bayesian update suggests using the prior parameter $\delta$ in the local client update, or the overall weight-decay. For $\rho$, we follow Ashman et al. (2022) and set it to $1/K$.

2. Second, by dividing the first line by $N_k$ we can recover $\bar{\ell}_k$ used in ADMM (see the first line in Eq. 2). This suggests that FedLap scales the $\|\boldsymbol{w} - \boldsymbol{w}_g\|^2$ term by $\delta/N_k$, matching with a practical choice in FedDyn's codebase which seems to work well empirically.

3. Third, the update of $\boldsymbol{w}_g$ in PVI does not have $\boldsymbol{w}_k$ in it or a division by $K$. This is due to updates that follow the form in Eq. 3, where we expect different $\hat{t}_k$ to already be automatically weighted: the Bayesian update does not suggest any additional weighting. Despite this difference, FedLap performs as well as the best-performing ADMM method (FedDyn) in our experiments.

FedDyn and FedLap have the same computation and communication cost. The communication cost is $P$ from clients to server (and server to clients), where $P$ is the number of parameters in the model. Each client $k$'s computation cost is dominated by $O(N_k P E)$, where $E$ is the number of epochs during optimisation. Computation cost at the global server is negligible, requiring simply adding together vectors of length $P$.

## 3.3 FEDLAP-COV: A NEW ADMM VARIANT WITH FLEXIBLE COVARIANCES

Our connection gives a direct way to design new ADMM variants to improve it: use a different candidate set $\mathcal{Q}$, that is, choose different posterior forms for the candidates $q(\boldsymbol{w})$. Here, we demonstrate this for multivariate Gaussian $q(\boldsymbol{w}) = \mathcal{N}(\boldsymbol{w}; \boldsymbol{m}, \boldsymbol{\Sigma})$ where, unlike the previous section, we aim to also estimate the covariance matrix $\boldsymbol{\Sigma}$. This extension leads to *two* dual variables, where the second dual variable acts as a preconditioner similar to the Newton update. We will often write the precision matrix $\boldsymbol{S} = \boldsymbol{\Sigma}^{-1}$ because it is directly connected to the Hessian (denoted by $\mathbf{H}$) which is more natural for a Newton-like update. In practice, we use a diagonal matrix because it scales better to large models, but our derivation is more general.

We make the following choices to derive the new variant which we will call FedLap-Cov,

$$q_g(\boldsymbol{w}) \propto \mathcal{N}(\boldsymbol{w}; \boldsymbol{w}_g, \boldsymbol{S}_g^{-1}), \qquad q_k(\boldsymbol{w}) \propto \mathcal{N}(\boldsymbol{w}; \boldsymbol{w}_k, \boldsymbol{S}_k^{-1}), \qquad p_0(\boldsymbol{w}) \propto \mathcal{N}(\boldsymbol{w}; 0, \mathbf{I}/\delta). \quad (10)$$

Similar to the isotropic Gaussian case, these choices imply that $\hat{t}_k(\boldsymbol{w})$ also takes a Gaussian form where we need to find a pair of dual variables, a vector $\boldsymbol{v}_k$ and a symmetric square matrix $\boldsymbol{V}_k$,

$$\hat{t}_k(\boldsymbol{w}) \propto e^{\boldsymbol{v}_k^\top \boldsymbol{w} - \frac{1}{2}\boldsymbol{w}^\top \boldsymbol{V}_k \boldsymbol{w}} \quad (11)$$

Again, the existence of $(\boldsymbol{v}_k^*, \boldsymbol{V}_k^*)$ is ensured due to Eq. 4 but we skip the details here.

Now, we can simply plug-in the forms of $(q_k, \hat{t}_k, q_g)$ into Eq. 6 to get the update. The main differences are, first, the inclusion of $\boldsymbol{V}_k$ in the client's dual term (highlighted with red) and, second, the preconditioning with $\boldsymbol{S}_g$ used in the server updates,

$$\text{Client updates:} \quad (\boldsymbol{w}_k, \boldsymbol{S}_k) \leftarrow \arg\min_{q \in \mathcal{Q}} \mathbb{E}_q[\ell_k(\boldsymbol{w})] + \mathbb{E}_q\left[\boldsymbol{v}_k^\top \boldsymbol{w} - \tfrac{1}{2}\boldsymbol{w}^\top \boldsymbol{V}_k \boldsymbol{w}\right] + \mathbb{D}_{KL}[q \,\|\, q_g],$$

$$\boldsymbol{v}_k \leftarrow \boldsymbol{v}_k + \rho\left(\boldsymbol{S}_k \boldsymbol{w}_k - \boldsymbol{S}_g \boldsymbol{w}_g\right) \quad \text{and} \quad \boldsymbol{V}_k \leftarrow \boldsymbol{V}_k + \rho\left(\boldsymbol{S}_k - \boldsymbol{S}_g\right),$$

$$\text{Server updates:} \quad \boldsymbol{w}_g \leftarrow \boldsymbol{S}_g^{-1} \sum_{k=1}^{K} \boldsymbol{v}_k, \qquad \text{where} \quad \boldsymbol{S}_g = \delta\mathbf{I} + \sum_{k=1}^{K} \boldsymbol{V}_k. \quad (12)$$

Similarly to before, the updates can be simplified by using a delta approximation to derive a Laplace variant. Because we want local $\boldsymbol{S}_k$ to also be updated, we can use a second-order delta approximation for any $q(\boldsymbol{w}) = \mathcal{N}(\boldsymbol{w}; \boldsymbol{m}, \boldsymbol{S}^{-1})$,

$$\mathbb{E}_q[\ell_k(\boldsymbol{w})] \approx \mathbb{E}_q\left[\ell_k(\boldsymbol{m}) + (\boldsymbol{w} - \boldsymbol{m})\nabla\ell_k(\boldsymbol{m}) + \tfrac{1}{2}(\boldsymbol{w} - \boldsymbol{m})^\top \mathbf{H}_k(\boldsymbol{m})(\boldsymbol{w} - \boldsymbol{m})\right]$$
$$= \ell_k(\boldsymbol{m}) + \tfrac{1}{2}\text{Tr}\left[\mathbf{H}_k(\boldsymbol{m})\boldsymbol{S}^{-1}\right], \quad (13)$$

where $\mathbf{H}_k(\boldsymbol{m})$ denotes the hessian of $\ell_k$ at $\boldsymbol{w} = \boldsymbol{m}$. The approximation decouples the optimisation over $\boldsymbol{m}_k$ and $\boldsymbol{S}_k$ at a client (we assume that $\mathbf{H}(\boldsymbol{m})$ does not depend on $\boldsymbol{m}$ to avoid requiring

higher-order derivatives), allowing us to write them as two separate updates,

FedLap-Cov client update: $\quad \boldsymbol{w}_k \leftarrow \arg\min_{\boldsymbol{w}} \ \ell_k(\boldsymbol{w}) + \boldsymbol{v}_k^\top \boldsymbol{w} - \frac{1}{2}\boldsymbol{w}^\top \boldsymbol{V}_k \boldsymbol{w} + \frac{1}{2}\|\boldsymbol{w} - \boldsymbol{w}_g\|_{\boldsymbol{S}_g}^2$

$$\boldsymbol{S}_k \leftarrow \mathbf{H}_k(\boldsymbol{w}_k) - \boldsymbol{V}_k + \boldsymbol{S}_g \qquad (14)$$

A full derivation is in App. B. Here, $\|\boldsymbol{w}\|_{\boldsymbol{S}}^2 = \boldsymbol{w}^\top \boldsymbol{S}\boldsymbol{w}$ denotes the Mahalanobis distance. The two updates are performed concurrently and we have used $\mathbf{H}_k(\boldsymbol{w}_k)$ at the most recent parameters $\boldsymbol{w}_k$. These two steps along with the update for $\boldsymbol{v}_k, \boldsymbol{V}_k, \boldsymbol{w}_g, \boldsymbol{S}_g$ in Eq. 12 constitute our new Laplace variant, which we call FedLap-Cov. In our experiments we use diagonal matrices ($\boldsymbol{V}_k, \mathbf{H}_k, \boldsymbol{S}$), which already shows improved performance.

FedLap-Cov improves several aspects of ADMM and FedLap. First, the update of $\boldsymbol{v}_k$ uses the parameters $\boldsymbol{S}_k \boldsymbol{w}_k$ and $\boldsymbol{S}_g \boldsymbol{w}_g$, weighted by their respective precisions. As a result, the uncertainty is naturally incorporated when summing $\boldsymbol{v}_k$ in the global updates to get $\boldsymbol{w}_g$. Second, the global updates use preconditioning through $\boldsymbol{S}_g$ which also incorporates uncertainty in the global variables. Third, $\boldsymbol{S}_g$ is an accumulation of all $\boldsymbol{V}_k$ which captures the difference between $\boldsymbol{S}_k$ and $\boldsymbol{S}_g$. Finally, the Euclidean distance is changed to a Mahalanobis distance. We expect these aspects to be useful when dealing with heterogeneity across clients. The covariances should help in automatic weighing of information, which can potentially improve both performance and convergence.

For a valid algorithm, we need to ensure that $\boldsymbol{S}_k$ and $\boldsymbol{S}_g$ remain positive-definite throughout. This can be ensured by rewriting the $\boldsymbol{V}_k$ update by substituting $\boldsymbol{S}_k$ from Eq. 14,

$$\boldsymbol{V}_k \leftarrow \boldsymbol{V}_k + \rho(\boldsymbol{S}_k - \boldsymbol{S}_g) = \boldsymbol{V}_k + \rho(\mathbf{H}_k(\boldsymbol{w}_k) - \boldsymbol{V}_k) = (1 - \rho)\boldsymbol{V}_k + \rho\mathbf{H}_k(\boldsymbol{w}_k).$$

Therefore, if the Hessian is positive semi-definite all $\boldsymbol{V}_k$ will be too, which will also imply both $\boldsymbol{S}_g$ and $\boldsymbol{S}_k$ remain positive definite. We can enforce this by using a Generalised Gauss-Newton approximation (Schraudolph, 2002; Martens, 2020) or by using recent variational algorithms that ensure positive definite covariances (Lin et al., 2020b; Shen et al., 2024). We provide the final implemented FedLap-Cov update in Eq. 21. Similarly to ADMM, PVI and FedLap, at convergence we have $\boldsymbol{w}_g^* = \boldsymbol{w}_k^*$ and $\boldsymbol{S}_g^* = \boldsymbol{S}_k^*$.

FedLap-Cov has communication cost $2P$ from clients to server (and server to clients), as we send both $v_k$ and $V_k$ (this is for a diagonal matrix, as in all our experiments; for a full-covariance structure the theoretical cost is $3P/2 + P^2/2$). Client $k$'s computation cost is $O(N_k P(E + 1))$ as we need an additional backward pass through the dataset to calculate $V_k$ in our implementation, which is a small additional cost. We could alternatively use more efficient implementations, such as online implementations, to reduce this cost (Daxberger et al., 2021; Shen et al., 2024). Overall, FedLap-Cov adds new improvements to FedLap by using a more flexible posterior distribution, and we see this empirically in Sec. 4.

### 3.4 FEDLAP-FUNC: IMPROVING BY INCLUDING FUNCTION-SPACE INFORMATION

In the previous section, we derived FedLap-Cov, which improves FedLap by including full covariance information. In this section, we derive FedLap-Func, which improves FedLap by including function-space information over unlabelled inputs. Specifically, we assume some inputs are available to both a local client and the global server, and send soft labels (predictions) over those inputs, along with the weights we send in FedLap. This additional information can be seen as improving the gradient reconstruction of each client's data compared to just a Gaussian weight-space approximation, thereby improving the quality of information transmitted between local clients and the global server (Khan and Swaroop, 2021; Daxberger et al., 2023).

We start by writing FedLap's update at the global server as an optimisation problem,

$$\boldsymbol{w}_g = \arg\min_{\boldsymbol{w}} \ \tfrac{1}{2}\delta\|\boldsymbol{w}\|^2 - \sum_{k=1}^{K} \underbrace{\log \hat{t}_k}_{\text{client } k\text{'s contribution}}, \qquad (15)$$

where we note that the solution to this problem (followed by calculating the Laplace covariance at $\boldsymbol{w}_g$) gives us FedLap-Cov's server update from Eq. 12, and it therefore looks very similar to Eq. 6.

Following ideas from continual learning and knowledge adaptation (Khan and Swaroop, 2021), we improve the contribution from client $k$ by also sending information in function-space, instead of only

sending in weight-space. We do this over a set of inputs $\mathcal{M}_k$ (where $\mathcal{M}_k$ can be different for each client, or shared). We derive a similar equation to Daxberger et al. (2023), but through a more general form which avoids a Taylor series expansion, and allows us to adapt the theory to federated learning. Each client contribution to the global server in Eq. 15 becomes (derivation in App. C.1),

$$\underbrace{\sum_{i \in \mathcal{M}_k} \ell(\hat{y}_i, \boldsymbol{w})}_{\text{function-space information}} \underbrace{- \tfrac{1}{2}(\boldsymbol{w} - \boldsymbol{w}_k)^\top [\mathbf{H}^{\mathcal{M}_k}](\boldsymbol{w} - \boldsymbol{w}_k)}_{\text{new weight-space term}} + \underbrace{\log \hat{t}_k(\boldsymbol{w})}_{\text{unchanged from before}} \;, \tag{16}$$

where $\ell(y_i, \boldsymbol{w})$ is the loss at parameters $\boldsymbol{w}$ with label $y_i$, $\hat{y}_i$ is the soft label (prediction) over an input $i \in \mathcal{M}_k$ using local client weights $\boldsymbol{w}_k$, and $\mathbf{H}^{\mathcal{M}_k}$ is the (approximate) Hessian of the loss over points in $\mathcal{M}_k$ at weights $\boldsymbol{w}_k$. We see that the new weight-space term is subtracted from the additional function-space information: this avoids double-counting the contribution from points $i \in \mathcal{M}_k$. Although using this in the optimisation at the global server in Eq. 15 seems to require sending additional weight-space information ($\boldsymbol{w}_k$ and $\mathbf{H}^{\mathcal{M}_k}$), the equations simplify, allowing us to still send only $\boldsymbol{V}_k$ and $\boldsymbol{v}_k$, except that compared to earlier, these now remove the contribution from points $i \in \mathcal{M}_k$. We additionally send $\hat{y}_i$ for $i \in \mathcal{M}_k$, which is a very small cost when the same inputs $\mathcal{M}_k$ are available to the clients and the server.

We can simplify the equations further by again approximating all the covariances to be equal to $\mathbf{I}/\delta$, like in FedLap. In App. C.2 we show that this gives,

$$\text{Client updates:} \quad \boldsymbol{w}_k \leftarrow \arg\min_{\boldsymbol{w}} \underbrace{\ell_k(\boldsymbol{w}) + \delta \boldsymbol{v}_k^\top \boldsymbol{w} + \tfrac{1}{2}\delta \|\boldsymbol{w} - \boldsymbol{w}_g\|^2}_{\text{Same as FedLap}}$$

$$\underbrace{- \sum_{i \in \mathcal{M}_k} \tau \ell(\hat{y}_i, \boldsymbol{w}) + \sum_{k'=1}^{K} \sum_{i \in \mathcal{M}_{k'}} \tau \ell(\hat{y}_{i,g}, \boldsymbol{w})}_{\text{Additional function-space terms}}, \tag{17}$$

$$\boldsymbol{v}_k \leftarrow \boldsymbol{v}_k + \rho(\boldsymbol{w}_k - \boldsymbol{w}_g),$$

$$\text{Server update:} \quad \boldsymbol{w}_g \leftarrow \arg\min_{\boldsymbol{w}} \sum_{k=1}^{K} \left[ \sum_{i \in \mathcal{M}_k} \tau \ell(\hat{y}_i, \boldsymbol{w}) - \delta \boldsymbol{v}_k^\top \boldsymbol{w} \right] + \tfrac{1}{2}\delta \|\boldsymbol{w}\|^2,$$

where $\hat{y}_i$ is the prediction over an input $i \in \mathcal{M}_k$ using the previous round's client model weights $\boldsymbol{w}_k$, $\hat{y}_{i,g}$ is the prediction using the global server weights $\boldsymbol{w}_g$, and we have introduced a hyperparameter $\tau$ that upweights the function-space contribution (we might want $\tau > 1$ when we have few points in $\mathcal{M}_k$). FedLap-Func keeps the properties of FedLap at a stationary point: $\boldsymbol{w}_g^* = \boldsymbol{w}_k^*$ (at a stationary point, all the function-space terms contribute zero gradient). We also see a nice property of the algorithm: FedLap-Func recovers FedLap when $\mathcal{M}_k = \emptyset$. FedLap-Func can therefore be seen as improving FedLap by introducing some points in function-space.

We note that our points in $\mathcal{M}_k$ do not have to be points from client's data $\mathcal{D}_k$, and can be publicly available (unlabelled) data, like in Federated Distillation (Li and Fedmd, 2019; Lin et al., 2020a; Seo et al., 2020; Wu et al., 2021). Our algorithm is related to these knowledge distillation approaches in Federated Distillation, except we also send weights, thereby connecting to non-distillation approaches. We hope future work can explore these connections further.

The communication cost of FedLap-Func per round is almost the same as FedLap: FedLap-Func has communication cost $P + C^2$, where $C$ is the number of classes ($C \ll P$), because we send predictions over $C$ points (one point per class in our experiments) and each prediction vector is of length $C$. On clients, the computation cost is now $O((N_k + M)PE)$, where $M$ is the number of points in memory ($M \ll N_k$, so the additional cost is small). On the server, the computation cost increases compared to FedLap: the server has to optimise an objective function instead of simply adding vectors. It is now $O(MPE_g)$, where $M$ is the number of points in memory and $E_g$ is the number of optimisation steps. This is less than the computation cost at clients (because $M \ll N_k$) and will typically not be a problem (typically the global server is large and has more compute than clients).

## 4 EXPERIMENTS

We run experiments on a variety of benchmarks, (i) on tabular data and image data, (ii) using logistic regression models, multi-layer perceptrons, and convolutional neural networks, (iii) at different client data heterogeneity levels, and (iv) with different numbers of clients. We focus on showing that (i) our VB-derived algorithm FedLap performs comparably to the best-performing ADMM algorithm FedDyn (despite having one fewer hyperparameter to tune: no local weight-decay), (ii) including full covariances in FedLap-Cov improves performance, and (iii) including function-space information in FedLap-Func improves performance. Hyperparameters and further details are in App. E.

**Models and datasets.** We learn a logistic regression model on two (binary classification) datasets: UCI Credit (Quinlan, 1987) and FLamby-Heart (Janosi et al., 1988; du Terrail et al., 2022). Details on the datasets are in App. E. We use the same heterogeneous split on UCI Credit as in previous work (Ashman et al., 2022), splitting the data into 10 clients (results with a homogeneous split of UCI Credit are in Table 4 in App. F). FLamby-Heart is a naturally-heterogeneous split consisting of data from 4 hospitals (du Terrail et al., 2022). We also train a 2-hidden layer perceptron (as in previous work (Acar et al., 2021; McMahan et al., 2016)) on MNIST (LeCun et al., 1998) and Fashion MNIST (FMNIST) (Xiao et al., 2017) with $K = 10$ clients. We use a random $10\%$ of the data in FMNIST to simulate having less data, and use the full MNIST dataset. We consider both a homogeneous split and a heterogeneous split. To test having more clients, we also use a heterogeneous split of (full) Fashion MNIST across 100 clients. Lastly, we train a convolutional neural network on CIFAR10 (Krizhevsky and Hinton, 2009), using the same CNN from previous work (Pan et al., 2020; Zenke et al., 2017). We split the data heterogeneously into $K = 10$ clients. For all heterogeneous splits, we sample Dirichlet distributions that decide how many points per class go into each client (details in App. E.1). Our sampling procedure usually gives 2 clients $50\%$ of all the data, and 6 clients have $90\%$ of the data. Within the clients, $60$-$95\%$ of client data is within 4 classes out of 10.

**Methods and hyperparameters.** We compare FedLap, FedLap-Cov and FedLap-Func with three baselines: FedAvg (McMahan et al., 2016), FedProx (Li et al., 2020) and FedDyn (Acar et al., 2021), which is the best-performing federated ADMM-style method (FedDyn performs much better than FedADMM). We fix the local batch size and local learning rate (using Adam (Kingma and Ba, 2015)) to be equal for every algorithm, and sweep over number of local epochs, the $\delta$ and $\alpha$ hyperparameters, for FedDyn the additional local weight-decay hyperparameter, and for FedLap-Func the additional $\tau$ hyperparameter. For FedLap-Func, we assume one randomly-selected point per class per client is available to the global server (two points for CIFAR10). This breaks the strictest requirement of not sharing any client data with the global server, however, this is very few points, and it might be reasonable to share a few random points having obtained prior permission. We do not report results of FedLap-Func on FLamby-Heart because of the sensitive nature of medical data.

We summarise results in Table 1, showing the average accuracy (across 3 random seeds for all datasets) after a certain number of communication rounds. We provide further results in more tables in App. F, where we also report the average number of communication rounds to specific accuracies. They all show similar conclusions.

**FedLap performs at least as well as FedDyn across datasets and splits, and both are better than FedAvg and FedProx.** We first compare FedLap with the other baselines. We see that FedLap performs at least as well as FedDyn on all datasets and heterogeneity levels, showing that FedLap is similarly strong to FedDyn, despite having one fewer hyperparameter to tune (see also results on a homogeneous split of UCI Credit in Table 4 in App. F, where FedLap performs better than FedDyn). We note that FedDyn's performance is very sensitive to the value of this additional weight-decay hyperparameter (we provide an example of this hyperparameter's importance in App. D.1). Additionally, we find that FedLap performs well with the optimal global weight-decay $\delta$ (we show an example of FedLap's improved performance over FedDyn due to this property in App. D.2).

**FedLap-Cov significantly improves upon FedLap.** We see that FedLap-Cov consistently improves upon FedLap (and other baseline methods including FedDyn). At the highest number of communication rounds in Table 1, FedLap-Cov's accuracy is higher than FedDyn's by 1.7%-5.9% on four of our dataset settings, and is only marginally worse on one (0.2% worse on CIFAR-10, which is within standard deviation; FedLap-Cov is also significantly better earlier in training). This empirically shows the benefit of including covariance information to improve upon ADMM-style methods.

| Dataset | Comm Round | FedAvg | FedProx | FedDyn | FedLap | FedLap -Cov | FedLap -Func |
|---|---|---|---|---|---|---|---|
| FLambyH | 10 | 72.6(1.1) | 77.5(1.5) | 76.9(0.5) | 77.5(1.1) | **79.2(1.8)** (↑2.3) | – |
| (heterog) | 20 | 77.6(0.6) | 77.7(0.5) | **78.1(0.6)** | 77.7(0.5) | **80.0(0.5)** (↑1.9) | – |
| UCI Credit | 10 | 77.0(1.3) | 73.6(2.2) | 73.5(1.0) | 77.1(2.3) | **80.4(0.4)** (↑7.1) | 77.1(1.1) |
| (heterog) | 25 | 76.3(3.5) | 75.0(4.1) | 79.9(2.6) | 80.7(1.4) | **84.2(2.0)** (↑4.3) | **83.6(2.7)** |
| | 50 | 78.6(3.3) | 79.3(0.9) | 83.5(2.2) | 83.5(1.8) | **85.2(1.6)** (↑1.7) | **85.8(2.0)** |
| MNIST | 10 | 97.9(0.0) | 97.9(0.1) | 98.0(0.1) | 98.0(0.0) | 98.0(0.1) (0.0) | 97.9(0.1) |
| (homog) | 20 | 98.1(0.1) | 98.2(0.1) | 98.1(0.1) | 98.2(0.0) | 98.2(0.1) (↑0.1) | 98.2(0.0) |
| MNIST | 10 | 97.5(0.1) | 97.5(0.1) | 97.0(0.2) | 97.4(0.1) | **97.6(0.1)** (↑0.6) | 97.5(0.1) |
| (heterog) | 25 | 97.9(0.2) | 97.9(0.1) | 97.8(0.0) | 98.0(0.1) | 98.0(0.2) (↑0.2) | 98.0(0.1) |
| | 50 | 98.0(0.1) | 98.0(0.1) | 98.0(0.1) | **98.2(0.1)** | 98.2(0.1) (↑0.2) | 98.0(0.1) |
| FMNIST | 10 | 72.3(0.4) | 72.2(0.3) | **75.3(0.8)** | 72.1(0.2) | 75.0(0.6) (↓0.3) | 73.7(0.7) |
| (homog) | 25 | 77.7(0.3) | 77.4(0.1) | 77.5(0.8) | 77.1(0.1) | **79.8(0.4)** (↑2.3) | **77.9(0.3)** |
| | 50 | 80.0(0.2) | **80.3(0.1)** | 78.2(0.5) | 80.2(0.1) | **81.8(0.1)** (↑2.6) | 80.0(0.2) |
| FMNIST | 10 | 70.4(0.9) | 69.9(0.4) | **73.0(0.6)** | 71.3(0.9) | **74.6(0.7)** (↑1.6) | 72.2(0.9) |
| (heterog) | 25 | 74.3(0.5) | 74.7(0.6) | 74.6(0.4) | 74.3(0.4) | **78.3(1.0)** (↑3.7) | 75.4(0.8) |
| | 50 | 76.0(0.7) | 76.9(0.9) | 74.6(0.5) | 77.6(0.7) | **80.5(0.6)** (↑5.9) | 78.1(0.7) |
| FMNIST | 10 | 73.9(0.3) | 73.3(0.7) | 75.7(0.4) | 73.9(0.3) | **76.9(0.7)** (↑1.2) | **76.9(0.8)** |
| (heterog) | 25 | 78.7(0.3) | 78.4(0.1) | **81.4(0.2)** | 79.4(0.2) | **81.3(0.2)** (↓0.1) | 79.7(0.4) |
| 100 clients | 50 | 81.8(0.3) | 81.6(0.1) | 82.2(0.3) | **82.4(0.3)** | **83.0(0.1)** (↑0.8) | 81.8(0.3) |
| CIFAR10 | 10 | 73.8(0.5) | 73.8(1.5) | 72.7(0.9) | 74.8(1.3) | **75.1(1.1)** (↑2.4) | **76.0(1.1)** |
| (heterog) | 25 | 75.0(0.5) | 75.0(0.9) | 77.4(0.6) | **78.0(1.2)** | 77.6(0.7) (↑0.2) | **78.5(1.0)** |
| | 50 | 75.1(0.3) | 75.4(0.8) | 79.4(0.4) | **79.5(1.4)** | 79.2(0.9) (↓0.2) | **79.5(1.4)** |

Table 1: The average accuracy (standard deviation in parentheses) over three runs reported after fixed numbers of communication rounds. We see that FedLap performs at least as well as FedDyn in all settings, while FedLap-Cov significantly improves over FedLap and FedDyn (except for homogeneous MNIST where all methods perform equally well): FedLap-Cov's improvement over FedDyn is reported next to FedLap-Cov's accuracy. FedLap-Func also improves performance (especially seen on CIFAR10 and heterogeneous FMNIST). For each run, we use the average accuracy over the previous 3 rounds to account for instabilities (maximum accuracy over previous 3 rounds in Table 2), and bold the top two performing algorithms (even if their standard deviations overlap with others).

**FedLap-Func improves upon FedLap.** We also see that FedLap-Func improves upon FedLap, especially in the heterogeneous splits of Fashion MNIST (10 clients) and CIFAR10. FedLap-Func often reaches better accuracies at earlier rounds, as it uses function-space information while the model is still far from optimal. We note that, in our experiments, we send function-space information over very few inputs (one randomly-chosen point per class per client, or two points with CIFAR10). Future work can look at using public unlabelled datasets to send function-space information over many more inputs, like in Federated Distillation. In App. E.5 we show that performance improves when we send function-space information over more datapoints (on FMNIST), as expected.

**We see the same conclusions with 100 clients instead of 10 clients.** We also run a heterogenous setting with 100 clients (instead of 10) to simulate having many more clients (results in Table 1). We do this on FashionMNIST (full dataset), using the same method to heterogenously split data across clients as with 10 clients. We see the same conclusions as with 10 clients: FedLap performs as well as FedDyn (and better than FedAvg and FedProx) despite having one fewer hyperparameter to tune, FedLap-Cov significantly improves performance, and FedLap-Func improves performance particularly earlier in training.

## 5 Conclusions and Future Work

In this paper, we provide new connections between two distinct and previously unrelated federated learning approaches based on ADMM and Variational Bayes (VB), respectively. Our key result shows that the dual variables in ADMM naturally emerge through the site parameters used in VB. We first show this for the isotropic Gaussian case and then extend it to multivariate Gaussians. The latter is used to derive a new variant of ADMM where learned covariances incorporate uncertainty and

are used as preconditioners. We also derive a new functional regularisation extension. Numerical experiments show that these improve performance.

This work is the first to show such connections of this kind. No prior work has shown the emergence of dual variables while estimating posterior distributions. The result is important because it enables new ways to combine the complementary strengths of ADMM and Bayes. We believe this to be especially useful for non-convex problems, such as those arising in deep learning.

There are many avenues for future work. We believe this connection holds beyond the federated learning setting, and we hope to explore this in the future. This could give rise to a new set of algorithms that combine duality and uncertainty to solve challenging problems in machine learning, optimisation, and other fields. There are also several extensions for the federated case. Our current experiments update all clients in every round of communication, and future experiments could relax this. We expect that our methods will still work well, just like algorithmically-related algorithms such as FedDyn. Future work can also analyse convergence rates of FedLap, following similar theoretical assumptions to those for FedDyn and FedADMM. We could also explore differentially-private versions of the algorithm, following differentially-private versions of PVI (Heikkilä et al., 2023). We also expect that our method will perform well in continual federated learning, using Bayesian continual learning techniques (Kirkpatrick et al., 2017; Pan et al., 2020; Titsias et al., 2020). Lastly, future work can explore connections between Federated Distillation and FedLap-Func.

## ACKNOWLEDGMENTS

This material is based upon work supported by the National Science Foundation under Grant No. IIS-2107391. Any opinions, findings, and conclusions or recommendations expressed in this material are those of the author(s) and do not necessarily reflect the views of the National Science Foundation. MEK is supported by the Bayes-Duality project, JST CREST Grant Number JPMJCR2112.

## AUTHOR CONTRIBUTIONS STATEMENT

List of authors: Siddharth Swaroop (SS), Mohammad Emtiyaz Khan (MEK), Finale Doshi-Velez (FDV).

SS led the project and conceived the original ideas with regular feedback from FDV and MEK. The emergence of dual variables through site parameter was hypothesised by MEK, and confirmed by SS. SS coded and ran all experiments. SS wrote a first draft of the paper with feedback from FDV, which was substantially revised by MEK with feedback from SS and FDV.

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

## A   MAKING FEDLAP MORE SIMILAR TO THE ADMM UPDATE

In this section, we re-write the FedLap update (Eq. 9) to look more similar to the ADMM update (Eq. 2). Specifically, we divide by $N_k$, and make the substitution $\tilde{\boldsymbol{v}}_k = \delta \boldsymbol{v}_k / N_k$, giving,

$$
\begin{aligned}
\text{Client updates:} \quad & \boldsymbol{w}_k \leftarrow \arg\min_{\boldsymbol{w}} \; \bar{\ell}_k(\boldsymbol{w}) + \tilde{\boldsymbol{v}}_k^\top \boldsymbol{w} + \tfrac{1}{2}(\delta/N_k)\|\boldsymbol{w} - \boldsymbol{w}_g\|^2 \\
& \tilde{\boldsymbol{v}}_k \leftarrow \tilde{\boldsymbol{v}}_k + \rho(\delta/N_k)(\boldsymbol{w}_k - \boldsymbol{w}_g) \\
\text{Server update:} \quad & \boldsymbol{w}_g \leftarrow \sum_{k=1}^{K} \frac{1}{(\delta/N_k)} \tilde{\boldsymbol{v}}_k.
\end{aligned}
\tag{18}
$$

## B   DERIVATION OF FEDLAP-COV UPDATES

Here, we give full derivation of the update given in Eq. 14. We use the definition of the KL divergence for two multivariate Gaussians $q = \mathcal{N}(\boldsymbol{w}; \boldsymbol{m}, \boldsymbol{S})$ and $q_g = \mathcal{N}(\boldsymbol{w}; \boldsymbol{w}_g, \boldsymbol{S}_g)$:

$$
\mathbb{D}_{KL}[q \,\|\, q_g] = \tfrac{1}{2}\|\boldsymbol{m} - \boldsymbol{w}_g\|_{\boldsymbol{S}_g}^2 + \tfrac{1}{2}\mathrm{Tr}\left(\boldsymbol{S}_g \boldsymbol{S}^{-1}\right) - \tfrac{1}{2}\log|\boldsymbol{S}_g \boldsymbol{S}^{-1}| + \text{const.}
$$

We plug this and Eq. 13 in the first update given in Eq. 12 to get the following,

$$
\begin{aligned}
(\boldsymbol{w}_k, \boldsymbol{S}_k) \leftarrow \arg\min_{\boldsymbol{m}, \boldsymbol{S}} \; & \ell_k(\boldsymbol{m}) + \tfrac{1}{2}\mathrm{Tr}\left[\mathbf{H}_k(\boldsymbol{m}_{\mathrm{old}})\boldsymbol{S}^{-1}\right] + \boldsymbol{v}_k^\top \boldsymbol{m} - \tfrac{1}{2}\mathrm{Tr}\left[\boldsymbol{V}_k\left(\boldsymbol{m}\boldsymbol{m}^\top + \boldsymbol{S}^{-1}\right)\right] \\
& + \tfrac{1}{2}\|\boldsymbol{m} - \boldsymbol{w}_g\|_{\boldsymbol{S}_g}^2 + \tfrac{1}{2}\mathrm{Tr}\left(\boldsymbol{S}_g \boldsymbol{S}^{-1}\right) - \tfrac{1}{2}\log|\boldsymbol{S}_g \boldsymbol{S}^{-1}|.
\end{aligned}
\tag{19}
$$

The $\boldsymbol{V}_k$ dual term above (the fourth term) is derived using the following identity,

$$
\mathbb{E}_q[\boldsymbol{w}^\top \boldsymbol{V}_k \boldsymbol{w}] = \mathbb{E}_q[\mathrm{Tr}\left(\boldsymbol{V}_k \boldsymbol{w}\boldsymbol{w}^\top\right)] = \mathrm{Tr}\left[\boldsymbol{V}_k \mathbb{E}_q\left(\boldsymbol{w}\boldsymbol{w}^\top\right)\right] = \mathrm{Tr}\left[\boldsymbol{V}_k\left(\boldsymbol{m}\boldsymbol{m}^\top + \boldsymbol{\Sigma}\right)\right].
$$

With this, the updates over $\boldsymbol{w}_k$ and $\boldsymbol{S}_k$ decouple into two different updates:

$$
\begin{aligned}
\boldsymbol{w}_k &\leftarrow \arg\min_{\boldsymbol{m}} \; \ell_k(\boldsymbol{m}) + \boldsymbol{v}_k^\top \boldsymbol{m} - \tfrac{1}{2}\boldsymbol{m}^\top \boldsymbol{V}_k \boldsymbol{m} + \tfrac{1}{2}\|\boldsymbol{m} - \boldsymbol{w}_g\|_{\boldsymbol{S}_g}^2, \\
\boldsymbol{S}_k &\leftarrow \arg\min_{\boldsymbol{S}} \; \tfrac{1}{2}\mathrm{Tr}\left[\mathbf{H}_k(\boldsymbol{m}_{\mathrm{old}})\boldsymbol{S}^{-1}\right] - \tfrac{1}{2}\mathrm{Tr}\left[\boldsymbol{V}_k \boldsymbol{S}^{-1}\right] + \tfrac{1}{2}\mathrm{Tr}\left(\boldsymbol{S}_g \boldsymbol{S}^{-1}\right) - \tfrac{1}{2}\log|\boldsymbol{S}_g \boldsymbol{S}^{-1}|.
\end{aligned}
\tag{20}
$$

By replacing $\boldsymbol{m}$ with $\boldsymbol{w}$, we recover the update for $\boldsymbol{w}_k$ in Eq. 14. The update of $\boldsymbol{S}_k$ has a closed form solution, as shown below by taking the derivative with respect to $\boldsymbol{S}^{-1}$ and setting it to 0,

$$
\mathbf{H}_k(\boldsymbol{m}_{\mathrm{old}}) - \boldsymbol{V}_k + \boldsymbol{S}_g - \boldsymbol{S}_k = 0 \quad \Longrightarrow \quad \boldsymbol{S}_k = \mathbf{H}_k(\boldsymbol{m}_{\mathrm{old}}) - \boldsymbol{V}_k + \boldsymbol{S}_g.
$$

In the main update, we set $\boldsymbol{m}_{\mathrm{old}}$ to be the most recent value of $\boldsymbol{m}_k$ before updating it. This ensures that the Hessian is estimated at the most recent parameters and so is the best estimate possible.

Using this derivation and the additional details in Sec. 3.3, we can write final FedLap-Cov update as,

$$
\begin{aligned}
\text{Client updates:} \quad & \boldsymbol{w}_k \leftarrow \arg\min_{\boldsymbol{w}} \; \ell_k(\boldsymbol{w}) + \boldsymbol{v}_k^\top \boldsymbol{w} - \tfrac{1}{2}\boldsymbol{w}^\top \boldsymbol{V}_k \boldsymbol{w} + \tfrac{1}{2}\|\boldsymbol{w} - \boldsymbol{w}_g\|_{\boldsymbol{S}_g}^2, \\
& \boldsymbol{S}_k \leftarrow \mathbf{H}_k(\boldsymbol{w}_k) - \boldsymbol{V}_k + \boldsymbol{S}_g, \\
& \boldsymbol{v}_k \leftarrow \boldsymbol{v}_k + \rho\left(\boldsymbol{S}_k \boldsymbol{w}_k - \boldsymbol{S}_g \boldsymbol{w}_g\right) \quad \text{and} \quad \boldsymbol{V}_k \leftarrow (1-\rho)\boldsymbol{V}_k + \rho\mathbf{H}_k(\boldsymbol{w}_k), \\
\text{Server updates:} \quad & \boldsymbol{w}_g \leftarrow \boldsymbol{S}_g^{-1} \sum_{k=1}^{K} \boldsymbol{v}_k, \qquad \text{where} \quad \boldsymbol{S}_g = \delta\mathbf{I} + \sum_{k=1}^{K} \boldsymbol{V}_k.
\end{aligned}
\tag{21}
$$

## C   DERIVATION OF FEDLAP-FUNC EQUATIONS

In this section we derive the FedLap-Func equations from Sec. 3.4.

## C.1 DERIVATION OF EQ. 16

At the global server, we add in the true loss over inputs in $\mathcal{M}_k$ (temporarily pretending as if we had true labels available), and then subtract the Gaussian approximate contribution over these points from Khan et al. (2019),

$$\sum_{i \in \mathcal{M}_k} \ell(y_i, \boldsymbol{w}) - \tfrac{1}{2}\Lambda_{i,k}(\tilde{y}_i - \boldsymbol{J}_i^\top \boldsymbol{w})^2, \tag{22}$$

where the second expression comes from Khan et al. (2019). They take a Laplace approximation of the posterior at $\boldsymbol{w}_k$, and show that this Laplace approximation is equal to the posterior distribution of the linear model $\tilde{y}_n = \boldsymbol{w}^\top \boldsymbol{J}_n + \epsilon_n$ after observing data $\tilde{\mathcal{D}} = \{\boldsymbol{J}_i, \tilde{y}_i\}_{i \in \mathcal{D}_k}$, where $\tilde{y}_i = \boldsymbol{w}_k^\top \boldsymbol{J}_i - \Lambda_{i,k}^{-1}(h_{i,k} - y_i)$, $\epsilon_n \sim \mathcal{N}(0, \Lambda_{n,k}^{-1})$, $\Lambda_{i,k}$ is the Hessian of the loss for point $i$ at parameters $\boldsymbol{w}_k$, and $\boldsymbol{J}_i$ is the Jacobian at input $i$ and weights $\boldsymbol{w}_k$ (this is equal to the input features in a Generalised Linear Model).

Substituting in the values of $\tilde{y}_i$, and noting that the Generalised Gauss Newton approximation to the Hessian is $\mathbf{H}^{\mathcal{M}_k} \approx \sum_{i \in \mathcal{M}_k} \Lambda_{i,k} \boldsymbol{J}_i \boldsymbol{J}_i^\top$, gives us that,

$$\sum_{i \in \mathcal{M}_k} \ell(y_i, \boldsymbol{w}) - \tfrac{1}{2}\Lambda_{i,k}(\tilde{y}_i - \boldsymbol{J}_i^\top \boldsymbol{w})^2 = \underbrace{\sum_{i \in \mathcal{M}_k} \ell(\hat{y}_i, \boldsymbol{w})}_{\text{function-space information}} \underbrace{- \tfrac{1}{2}(\boldsymbol{w} - \boldsymbol{w}_k)^\top [\mathbf{H}^{\mathcal{M}_k}](\boldsymbol{w} - \boldsymbol{w}_k)}_{\text{new weight-space term}},$$

$$\tag{23}$$

where $\hat{y}_i$ are soft labels (predictions) over inputs $i \in \mathcal{M}_k$ using local client weights $\boldsymbol{w}_k$, and $\mathbf{H}^{\mathcal{M}_k}$ is the (approximate) Hessian of the loss over points in $\mathcal{M}_k$ at weights $\boldsymbol{w}_k$. Adding this to Eq. 15 completes the derivation of Eq. 16.

## C.2 DERIVING THE FEDLAP-FUNC ALGORITHM

We derive the FedLap-Func equations (Eq. 17) from Eqs. 15 and 16. We note that we can derive a combined version of FedLap-Cov and FedLap-Func, but for simplicity we will focus on the non-covariance case.

We start with our approximation that all covariances are set to $\delta\mathbf{I}$, just like in FedLap: $\boldsymbol{S}_g = \boldsymbol{S}_k = \delta\mathbf{I}$. We note that normally, $\boldsymbol{S} = \mathbf{H} + \delta\mathbf{I}$, where $\mathbf{H}$ is the Hessian over points (eg the Generalised Gauss-Newton approximation). By making this approximation, we are ignoring this second-order information, setting all $\mathbf{H} = 0$. This means the contribution from client $k$ to the server (Eq. 16) becomes,

$$\underbrace{\sum_{i \in \mathcal{M}_k} \ell(\hat{y}_i, \boldsymbol{w})}_{\text{function-space information}} + \delta \boldsymbol{v}_k^\top \boldsymbol{w}, \tag{24}$$

where the new weight-space term has been removed as $\mathbf{H}^{\mathcal{M}_k} = 0$. We have simply added the function-space information, where $\hat{y}_i$ is the prediction over inputs $i \in \mathcal{M}_k$ at model parameters $\boldsymbol{w}_k$.

We include this function-space term in both the client update for $\boldsymbol{w}_k$ and the global server update for $\boldsymbol{w}_g$. We illustrate this with the global server update. Using App. C.1 gives us,

$$\underbrace{\sum_{k=1}^{K} \sum_{i \in \mathcal{M}_k} \ell(\hat{y}_{i,g}, \boldsymbol{w})}_{\text{function-space information}} \underbrace{- \tfrac{1}{2}(\boldsymbol{w} - \boldsymbol{w}_g)^\top [\mathbf{H}_g^{\mathcal{M}_k}](\boldsymbol{w} - \boldsymbol{w}_g)}_{\text{new weight-space term}} + \underbrace{\tfrac{1}{2}(\boldsymbol{w} - \boldsymbol{w}_g)^\top \boldsymbol{S}_g(\boldsymbol{w} - \boldsymbol{w}_g) - \delta\mathbf{I}}_{\text{unchanged from before}}, \tag{25}$$

where $\hat{y}_{i,g}$ is the prediction over inputs $i \in \mathcal{M}_k$ using model parameters $\boldsymbol{w}_g$ from the global server, and $\mathbf{H}_g^{\mathcal{M}_k}$ is the Hessian over points in $\mathcal{M}_k$ at parameter $\boldsymbol{w}_g$. In practice, we can approximate this Hessian with a Gauss-Newton approximation, and can either recalculate it at the global server before sending back to local clients, or approximate it as the Hessian sent from the client (calculated at $\boldsymbol{w}_k$), $\mathbf{H}_g^{\mathcal{M}_k} \approx \mathbf{H}^{\mathcal{M}_k}$.

Eq. 25 is simpler than Eq. 16, as the two weight-space terms have the same $(\boldsymbol{w} - \boldsymbol{w}_g)$ quadratic form. Therefore we can straightforwardly consider this as changing the global server precision $\boldsymbol{S}_g$ to remove the contribution from $i \in \mathcal{M}_k, \forall k$. When we take the approximation that all $\boldsymbol{S} = \delta \mathbf{I}$, we again set the Hessians to all be $0$, and this simplifies the equation further. Plugging this into the first client update in Eq. 9, and multiplying all function-space terms by a hyperparameter $\tau$, gives us our final FedLap-Func algorithm.

## D    BENEFITS OF FEDLAP OVER FEDDYN AND FEDADMM

In this section, we show two simple experiments showcasing why FedLap has better properties than FedDyn and FedADMM. First, in App. D.1 we show on a homogoneous MNIST split that FedDyn requires an additional weight-decay term in the local loss during training in order for it to perform well, unlike FedLap. Second, in App. D.2 we show how FedLap targets a better global loss (with weight-decay incorporated already), and gets closer as the number of communication rounds increases, unlike FedDyn and FedADMM, which target a worse global optimum (with no weight-decay in the global loss).

### D.1    IMPORTANCE OF LOCAL WEIGHT-DECAY IN FEDDYN

In this section, we show the importance of the additional weight-decay parameter to FedDyn's performance on a simple example (homogeneous MNIST), a hyperparameter that FedLap removes. Fig. 1 shows the results. We first see that FedDyn (with optimal weight-decay setting) and FedLap perform similarly. However, FedDyn's performance is highly dependent on the setting of its weight-decay hyperparameter: changing it by an order of magnitude in either direction significantly degrades performance. Removing it entirely results in catastrophic performance after the first 7 communication rounds. In contrast, changing the $\delta$ hyperparameter in FedLap (or FedDyn) by an order of magnitude does not reduce performance significantly in this setting.

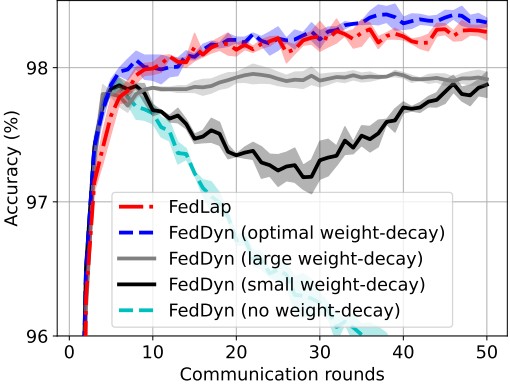

Figure 1: FedDyn's performance is sensitive to the additional weight-decay hyperparameter that it has compared to FedLap. When set an order of magnitude too small or too large, performance drops significantly. When there is no weight-decay, performance gets worse over rounds. Results are mean and standard deviation over three seeds on homogeneous MNIST (10 clients).

### D.2    FEDLAP TARGETS A BETTER GLOBAL LOSS THAN FEDDYN/FEDADMM

In this section, we present a simple setting showcasing the potential benefits of FedLap over FedDyn (and FedADMM). We take the UCI Credit dataset (a binary classification task), split the data equally (and homogeneously) into just two clients, and train a logistic regression classifier, ensuring we train until convergence both locally (full-batch, Adam learning rate 0.001, 1000 local epochs/iterations) and for a large number of communication rounds (1000 communication rounds). FedADMM is always slower and worse-performing than FedDyn, so we only consider FedDyn. Note that in this

section, we do not include weight-decay in the local optimisation of FedDyn, because FedDyn's theory does not include this, and this setting is very simple.

First, we note that ideal global performance (of $w_g^*$) is better with $\delta = 1$ (giving a test nll of 0.337) than $\delta = 0$ (test nll 0.420). Therefore the $w_g^*$ that FedLap targets is already a better target than that of FedDyn (and FedADMM), which targets a global optimum with $\delta = 0$.

Secondly, FedLap is quicker to target its ideal global parameter than FedDyn. FedLap reaches within 1% of its ideal $w^*$ train loss within just 2 communication rounds, and continues getting closer as we train for longer (up to the 1000 rounds). This happens even when setting different values of $\delta$ (meaning that FedLap might be targeting unideal values of $w_g^*$).

On the other hand, how close FedDyn gets to its own (worse) ideal train nll depends on the value of its hyperparameter $\alpha$: the best value takes 78 rounds to get to within 1% of its optimal train nll. Larger values of the hyperparameter do not reach within 1% within 1000 rounds: they lead to quick improvement earlier in training, but then performance (and train nll) gets worse over time.

# E  ADDITIONAL EXPERIMENTAL DETAILS

In this section we provide further details on experimental setup, including hyperparameter values.

For FedLap-Func, we set a datapoint-specific value of $\tau$, making it depend on the number of datapoints from that class in the client: for datapoint $i$ belonging to class $c$, we have $\tau_i = \tau_f \frac{N_{k,c}}{M_{k,c}}$, where $N_{k,c}$ is the number of datapoints in the class $c$ in client $k$, and $M_{k,c}$ is the number of datapoints in class $c$ in memory $\mathcal{M}_k$ (usually 1). We then perform a hyperparameter sweep over $\tau_f$. In this way, we multiply the contribution of each datapoint by how many datapoints it is representing from class $c$ in client $k$.

## E.1  HETEROGENEOUS SAMPLING PROCEDURE FOR MNIST, FASHION MNIST AND CIFAR10

Similar to previous work, we use two Dirichlet distributions to sample heterogeneous data splits. We first sample from $\mathrm{Dir}(\boldsymbol{\alpha}_1)$, where $\boldsymbol{\alpha}_1$ is of length number of clients $K$, to sample client sizes. We then sample a separate $\mathrm{Dir}(\boldsymbol{\alpha}_2)$ within each client, where $\boldsymbol{\alpha}_2$ is of length number of classes, to sample class distribution within a client. We then multiply the sampled class distribution within each client by the sampled client size. When assigning datapoints from a specific class to clients, we normalise these values across clients, so that all datapoints are assigned between the clients.

We use $\boldsymbol{\alpha}_1 = 1$ and $\boldsymbol{\alpha}_2 = 0.5$ in our heterogeneous splits on MNIST, Fashion MNIST and CIFAR10. Our sampling procedure usually gives 2 clients 50% of all the data, and 6 clients have 90% of the data. Within the clients, 60-95% of client data is within 4 classes. We use the same $\boldsymbol{\alpha}$ values for the 100-client case.

## E.2  UCI CREDIT

In UCI Credit, the binary classification task is to predict whether individuals default on payments, and there are a total of 520 training points (with 45% positive labels), with an accuracy of 90% (when global weight-decay is not zero). We use the heterogeneous split from Ashman et al. (2022) in the main text: data is split into $K = 10$ clients, 5 of which have 36 datapoints each (positive label rate of 6%), and the other 5 have 67 datapoints each (positive label rate of 66%). We provide results for a homogeneous split over $K = 10$ clients in Table 4, where we see similar conclusions (there is an even bigger improvement of FedLap and FedLap-related methods).

We set local Adam learning rate at $10^{-3}$, and minibatch size to 4 (ensuring there is gradient noise always). All methods have a hyperparameter sweep over number of local epochs = $[5, 10, 20]$. We perform a sweep over the $\alpha$ or $\delta$ hyperparameter for FedProx, FedLap, FedLap-Cov and FedLap-Func over $[10, 1, 10^{-1}]$. For FedDyn the sweep is over $\alpha = [10^{-3}, 10^{-4}, 10^{-5}]$, and local weight-decay sweep is over $[10^{-4}, 10^{-5}, 10^{-6}]$. For FedLap and FedLap-Func, we set damping factor $\rho = N_k/N$, and for FedLap-Cov, $\rho = 1/K$. For FedLap-Func, we set $\tau = 1$, and run global server optimisation for 5000 iterations with a learning rate of 0.001. Each run took up to 4 minutes on a standard laptop CPU.

### E.3 FLAMBY-HEART

FLamby-Heart is a naturally-heterogeneous split, proposed by du Terrail et al. (2022) as a federated learning benchmark, and consisting of data from 4 hospitals to predict whether a patient has heart disease (binary classification). There are 486 training examples.

We set batch size to 4 (like in du Terrail et al. (2022)), but use the Adam optimiser locally. We perform hyperparameter sweeps over number of local epochs = $[1, 5, 10]$, and (Adam) local learning rate = $[10^{-3}, 10^{-2}]$. For FedProx, FedLap, and FedLap-Cov, we perform a sweep over $\delta$ (or $\alpha$) = $[100, 10, 1, 10^{-1}, 10^{-2}]$. For FedDyn, this is over $\alpha = [10, 1, 10^{-1}, 10^{-2}, 10^{-3}]$, and FedDyn's local weight-decay sweep is over $[0, 10^{-2}, 10^{-3}, 10^{-4}]$. For FedLap, we set damping factor $\rho = N_k/N$, and for FedLap-Cov, $\rho = 1/K$. Each run took less than 1 minute on a standard laptop CPU.

Note that we do not report results of FedLap-Func as FLamby-Heart has sensitive data (medical data), where it is not likely to be reasonable to share this data with a global server.

### E.4 MNIST

We use the full MNIST dataset. Batch size is set to 32. We train a two-hidden layer multi-layer perceptron, with 200 hidden units in the first layer, and 100 units in the second layer (the model gets 98.3% accuracy on all data).

For all methods, we perform hyperparameter sweeps over number of local epochs = $[1, 5, 10]$, and (Adam) local learning rate = $[10^{-3}, 10^{-2}]$. For FedProx, FedLap, FedLap-Cov and FedLap-Func, we perform a sweep over $\delta$ (or $\alpha$) = $[1, 10^{-1}, 10^{-2}]$. For FedDyn, this is over $\alpha = [10^{-4}, 10^{-5}, 10^{-6}]$, and FedDyn's local weight-decay sweep is over $[10^{-4}, 10^{-5}, 10^{-6}]$. For FedLap and FedLap-Func, we set damping factor $\rho = N_k/N$, and for FedLap-Cov, $\rho = 1/K$. For FedLap-Func, we sweep over $\tau_f = [1, 10^{-1}, 10^{-2}]$, and run global server optimisation for 3000 iterations with a learning rate of 0.0005. Each run took up to 2 hours on a standard laptop CPU.

### E.5 FASHION MNIST

We use a random 10% of Fashion MNIST every random seed, and average across (the same) three random seeds. Batch size is set to 32. We train a two-hidden layer multi-layer perceptron, with 200 hidden units in the first layer, and 100 units in the second layer (the model gets 85% accuracy). Local Adam learning rate is set to $10^{-3}$.

For all methods, we perform hyperparameter sweeps over number of local epochs = $[1, 5, 10]$. For FedProx, FedLap, FedLap-Cov and FedLap-Func, we perform a sweep over $\delta$ (or $\alpha$) = $[1, 10^{-1}, 10^{-2}, 10^{-3}, 10^{-4}]$. For FedDyn, this is over $\alpha = [10^{-2}, 10^{-3}, 10^{-4}, 10^{-5}, 10^{-6}]$, and FedDyn's local weight-decay sweep is over $[10^{-4}, 10^{-5}, 10^{-6}]$. For FedLap and FedLap-Func, we set damping factor $\rho = N_k/N$, and for FedLap-Cov, $\rho = 1/K$. For FedLap-Func, we sweep over $\tau_f = [1, 10^{-1}, 10^{-2}, 10^{-3}, 10^{-4}]$, and run global server optimisation for 5000 iterations with a learning rate of 0.001. Each run took up to 1 hour 20 minutes on a standard laptop CPU.

**FedLap-Func: sending information over more datapoints.** If sending information over 2 points per class per client (instead of 1), FedLap-Func improves performance. In the homogeneous setting, mean accuracies (compare with Table 1) are 74.0(0.7) at round 10, 77.8(0.4) at round 25, 80.1(0.4) at round 50. Max accuracies (compare with Table 2) are 74.4(0.6) at round 10, 78.2(0.2) at round 25, 80.3(0.4) at round 50. 75% accuracy is reached in 12(2) communication rounds, and 78% in 25(2) rounds (compare with Table 3). In the heterogeneous setting, mean accuracies (compare with Table 1) are 73.2(0.1) at round 10, 76.1(0.6) at round 25, 78.4(0.2) at round 50. Max accuracies (compare with Table 2) are 73.7(0.3) at round 10, 76.7(0.7) at round 25, 78.8(0.2) at round 50. 75% accuracy is reached in 16(2) communication rounds, and 78% in 37(5) rounds (compare with Table 3).

### E.6 CIFAR-10

Batch size is set to 64. We set Adam learning rate = $10^{-3}$ (note that $10^{-2}$ fails to learn anything).

For all methods, we perform hyperparameter sweeps over number of local epochs = $[5, 10, 20]$. For FedProx, we perform a sweep over $\alpha = [1, 10^{-1}, 10^{-2}, 10^{-3}]$. For FedLap, FedLap-Cov

and FedLap-Func, we perform a sweep over $\delta = [1, 10^{-1}, 10^{-2}]$. For FedDyn, this is over $\alpha = [10^{-3}, 10^{-4}, 10^{-5}]$, and FedDyn's local weight-decay sweep is over $[10^{-4}, 10^{-5}, 10^{-6}]$.

For FedLap and FedLap-Func, damping $\rho$ was $N_k/N$ for first 10 rounds of training, then $1/K$ after that, and gradient clipping was used to stabilise training. For FedLap-Func, we sweep over $\tau_f = [10^{-2}, 10^{-4}, 10^{-6}]$, and run global server optimisation for 500 iterations with learning rate 0.0005. We assume that each client has 2 randomly-selected points per class shared with the global server (all other dataset/benchmark settings have 1 point per class per client). Each run took up to 10 hours on an A100 GPU.

# F  ADDITIONAL EXPERIMENTAL RESULTS

In this section we provide some more results from our experiments. We see the same conclusions / results as from Table 1 in the main text.

Table 2 reports the maximum accuracy over the previous three communication rounds (instead of the mean accuracy), and averages this across 3 runs. We see the same results and conclusions as from Table 1: FedLap performs similarly to FedDyn across all dataset and heterogeneity settings, and FedLap-Cov and FedLap-Func improve upon this. Table 3 shows the number of communication rounds to a desired accuracy. Again, we see the same takeaways / conclusions.

We provide additional results on a homogeneous UCI Credit split over $K = 10$ clients in Table 4. Here, FedLap performs even better than FedDyn consistently, because it is targeting a better global loss (with non-zero weight-decay).

In Table 5 we report average test negative log-likelihood results (instead of accuracy) at the end of training. We see that on average, FedLap performs at least as well as FedDyn (sometimes significantly outperforming FedDyn, such as on FMNIST (homog), FMNIST (heterog) and CIFAR10). FedLap-Cov and FedLap-Func improve even further, as might be expected due to the Bayesian interpretation.

| **Dataset** | Comm Round | FedAvg | FedProx | FedDyn | FedLap | FedLap -Cov | FedLap -Func |
|---|---|---|---|---|---|---|---|
| MNIST homog | 10 | 98.0(0.0) | 98.0(0.1) | **98.1(0.1)** | 98.0(0.0) | **98.1(0.1)** | 98.0(0.1) |
| | 20 | 98.2(0.0) | **98.3(0.1)** | 98.2(0.0) | 98.2(0.0) | **98.3(0.0)** | 98.2(0.0) |
| heterog | 10 | 97.6(0.1) | 97.6(0.1) | 97.1(0.2) | 97.5(0.1) | **97.7(0.2)** | **97.7(0.1)** |
| | 25 | 97.9(0.1) | 97.9(0.1) | 98.0(0.1) | 98.0(0.1) | 98.0(0.2) | **98.1(0.1)** |
| | 50 | 98.0(0.1) | 98.1(0.1) | 98.1(0.0) | **98.2(0.1)** | **98.2(0.1)** | 98.1(0.1) |
| FMNIST homog | 10 | 73.6(1.0) | 73.4(0.3) | **75.8(0.4)** | 73.1(0.2) | **76.0(0.2)** | 74.5(0.5) |
| | 25 | 78.2(0.3) | 77.9(0.1) | 78.0(0.8) | 77.5(0.2) | **80.2(0.3)** | 78.6(0.5) |
| | 50 | 80.3(0.1) | **80.5(0.1)** | 78.5(0.3) | 80.4(0.1) | **82.0(0.2)** | 80.4(0.1) |
| heterog | 10 | 71.3(0.7) | 71.2(0.7) | **74.0(0.3)** | 72.2(0.6) | **75.4(0.9)** | 72.7(0.9) |
| | 25 | 74.7(0.5) | 75.1(0.3) | 75.1(0.8) | 74.9(0.8) | **78.6(1.0)** | 75.7(0.8) |
| | 50 | 76.4(0.5) | 77.4(0.8) | 75.0(0.6) | 78.1(0.6) | **80.9(0.5)** | 78.5(0.7) |
| CIFAR10 heterog | 10 | 74.0(0.5) | 74.2(1.4) | 73.2(0.8) | 75.3(1.1) | **75.4(1.1)** | **76.6(1.5)** |
| | 25 | 75.1(0.5) | 75.1(0.8) | 77.6(0.7) | **78.2(1.1)** | 77.8(0.8) | **78.6(1.0)** |
| | 50 | 75.3(0.3) | 75.6(0.7) | 79.6(0.4) | **79.7(1.3)** | 79.3(0.8) | **79.6(1.3)** |

Table 2: Mean accuracy (standard deviation in parentheses) over three runs (except CIFAR-10), after a fixed number of communication rounds. For each run, we use the maximum accuracy over the previous 3 rounds (mean accuracy over previous 3 rounds in Table 1). We bold the top two performing algorithms (even if their standard deviations overlap with others). We see that FedLap performs at least as good as FedDyn in all settings (with a bigger difference when the global model performs better with a non-zero weight decay, like in UCI Credit). FedLap-Cov significantly improves upon FedLap in most settings (except for homogeneous MNIST, where all methods perform equally well), and FedLap-Func also improves performance (especially seen on CIFAR10).

| Dataset | Acc (%) | FedAvg | FedProx | FedDyn | FedLap | FedLap -Cov | FedLap -Func |
|---|---|---|---|---|---|---|---|
| MNIST homog | 97.5 | 6(1) | 6(1) | **4(1)** | 5(1) | **4(1)** | 6(1) |
| heterog | 97.5 | 10(2) | 10(2) | 18(3) | 11(1) | **9(1)** | **9(1)** |
| FMNIST homog | 75 | 15(1) | 15(2) | **9(1)** | 15(1) | **9(1)** | 12(2) |
| | 78 | 25(1) | 26(3) | **17(3)** | 28(1) | **15(1)** | 24(1) |
| heterog | 75 | 26(4) | 22(5) | 24(7) | 24(4) | **11(3)** | **20(2)** |
| | 78 | – | – | – | – | **22(6)** | – |
| CIFAR10 heterog | 72 | 6(1) | 6(3) | 8(1) | 7(1) | 6(1) | **5(1)** |
| | 75 | – | – | 14(2) | 10(3) | 10(3) | **8(2)** |
| | 78 | – | – | **30(5)** | 32(11) | 32(11) | **22(10)** |

Table 3: Mean number of communication rounds (standard deviation in parentheses) over three runs, to reach desired accuracy. If every run does not reach desired accuracy within 50 communication rounds, no number is reported. We bold the top two performing algorithms (even if their standard deviations overlap with others). We see that FedLap performs similarly to FedDyn in all settings. FedLap-Cov significantly improves upon FedLap in most settings, and FedLap-Func also improves performance (especially seen on CIFAR10).

| Dataset | Comm Round | FedAvg | FedProx | FedDyn | FedLap | FedLap -Cov |
|---|---|---|---|---|---|---|
| UCI Credit | 10 | 81.4(0.6) | 82.1(0.5) | 81.8(1.4) | **83.6(0.4)** | **84.4(0.5)** |
| (homog) | 25 | 84.6(1.1) | 84.2(0.4) | 86.3(1.5) | **87.0(0.4)** | **86.5(1.0)** |
| | 50 | 86.8(0.6) | 87.4(0.1) | 86.9(0.5) | **88.0(0.6)** | **87.7(0.7)** |
| | 100 | **88.7(0.2)** | 88.5(0.6) | 87.9(0.3) | 88.2(0.2) | **89.1(0.7)** |

Table 4: Mean accuracy (standard deviation in parentheses) over three runs, after a fixed number of communication rounds, on a homogoneous split of UCI Credit into $K = 10$ clients. For each run, we use the average accuracy over the previous 3 rounds to account for instabilities, and bold the top two performing algorithms (even if their standard deviations overlap with others). We see that FedLap and FedLap-Cov perform the best.

| Dataset | Comm Round | FedAvg | FedProx | FedDyn | FedLap | FedLap -Cov | FedLap -Func |
|---|---|---|---|---|---|---|---|
| MNIST (homog) | 20 | 0.11(0.01) | 0.07(0.04) | **0.06(0.00)** | 0.09(0.00) | 0.09(0.01)(↑0.03) | 0.07(0.00) |
| MNIST (heterog) | 50 | 0.13(0.01) | 0.12(0.01) | 0.07(0.01) | **0.06(0.00)** | 0.07(0.00)(0.00) | 0.06(0.00) |
| FMNIST (homog) | 50 | 0.56(0.01) | 0.56(0.01) | 0.63(0.01) | 0.57(0.01) | 0.56(0.00)(↓0.07) | 0.57(0.00) |
| FMNIST (heterog) | 50 | 0.67(0.04) | 0.66(0.03) | 0.77(0.05) | 0.65(0.05) | **0.60(0.04)(↓0.17)** | **0.62(0.02)** |
| FMNIST (heterog) 100 clients | 50 | 0.53(0.01) | 0.53(0.01) | **0.51(0.01)** | 0.53(0.01) | **0.49(0.01)(↓0.02)** | 0.52(0.01) |
| CIFAR10 (heterog) | 50 | 1.5(0.1) | 1.5(0.1) | 0.7(0.0) | 0.6(0.0) | 0.6(0.0)(↓0.1) | 0.6(0.0) |

Table 5: The average validation negative log-likelihood (standard deviation in parentheses) over three runs reported after a few fixed number of communication rounds. Lower is better. We see that on average, FedLap performs at least as well as FedDyn (sometimes significantly outperforming FedDyn, such as on FMNIST (homog), FMNIST (heterog) and CIFAR10). FedLap-Cov and FedLap-Func improve even further, as might be expected due to the Bayesian interpretation.

