# OpenReview forum: "Connecting Federated ADMM to Bayes"
_ICLR.cc/2025/Conference — ICLR 2025 Poster_

### Official Review · Reviewer_kAKo · 2024-11-01

**Soundness:** 3
**Presentation:** 3
**Contribution:** 2
**Rating:** 6
**Confidence:** 4

**Summary:**

This paper establishes the connection between federate learning and variational Bayes through the "site" parameters in partition variational inference (PVI). For particular choice of variational distribution in VB, PVI could (almost) be identified as an Alternative Direction Methods of Multiplier (ADMM), hence named FedLap. Base on such connection, the authors propose two improvements of FedLap (ADMM) by considering full covariances (FedLap-Cov) and function space (FedLap-Func) information coming from VB literature. Then multiple benchmark tests are carried out to demonstrate the superior performance of the proposed improvements compared with vanilla FedLap and competitive federate learning algorithms including FedAvg, FedProx, and FedDyn.

**Strengths:**

A nice connection is drawn between federate learning and variational Bayes. Two algorithms improving upon ADMM are proposed based the newly discovered connection. There is good potential in proposed methods, both numerically and methodologically.

**Weaknesses:**

FedLap-Func only provides marginal improvements over FedLap-Cov and sometimes even makes it worse. Did you explore other problems outside the benchmarks to seek possibly more appropriate applications?

**Questions:**

* Could you please include a paper outline at the end of Section 1?

* Could you bring the definition of $\hat t^*_k=\frac{q_k}{q_g}$ from (5) to (4) where it appears for the first time? I guessed $\hat t^*_k=\frac{q_k}{p_0}$ while at equation (4).

* `FedLap` seemingly indicates the Laplace variant of the PVI. But it is still Gaussian used as the variational distribution. Why is it called "Lap"? Would you please elaborate?

* FedLap-Func only provides marginal improvements over FedLap-Cov and sometimes even makes it worse. Any explanation?

---

> ### Author Response · Authors · 2024-11-20
> **Author response**
>
> We thank the reviewer for their comments and review. We are glad that the reviewer appreciates the connection between ADMM and Bayes, and appreciates the numerical and methodological parts of the new methods. We respond to the reviewer’s questions individually below. We have updated our paper on openreview, with changes highlighted in blue.
>
> **Q1: “FedLap-Func only provides marginal improvements over FedLap-Cov and sometimes even makes it worse.”**
> We expect FedLap-Func to improve upon FedLap, not also improve upon FedLap-Cov. This is because FedLap-Func is an orthogonal way to improve FedLap compared to FedLap-Cov: FedLap-Func adds function-space information only, while FedLap-Cov adds covariance information. It is possible to combine the two (use both covariance information and function-space information in a single algorithm), but we do not test this in the experiments. We did not explore problems outside the benchmarks listed, because already FedLap-Func is outperforming FedLap and other existing baselines.
>
> **Q2: “Include a paper outline at the end of Section 1.”**
> Thanks, we have done so in the new version of the paper.
>
> **Q3: “Could you bring the definition of $\hat{t}^\*_k$ from (5) to (4)”.**
> Note that $\hat{t}^*_k$ used in Eq 4 are different than $\hat{t}_k$ used in Eq 5, therefore it is not possible to move them or use them interchangeably. Eq 5 gives a way to estimate the optimal sites in Eq 4, that is, as the PVI algorithm progresses, $\hat{t}_k$ converges to $\hat{t}^*_k$.
>
> **Q4: “FedLap seemingly indicates the Laplace variant of the PVI. But it is still Gaussian used as the variational distribution. Why is it called "Lap"?”**
> Good question. FedLap is indeed a Laplace variant obtained through the variational PVI algorithm. Essentially, by taking a delta approximation in the PVI, we recover the Laplace variant. We also show how FedLap’s fixed points coincide with the Laplace solution of the global problem (old version lines 201-204, 274-275 and 341-342; new version lines 210-212, 291-292, 355-356).

---

> > ### Comment · Reviewer_kAKo · 2024-11-27
> >
> > Thanks to the authors for their clarification. I have raised my score.

---

> > > ### Author Response · Authors · 2024-11-27
> > > **Author response**
> > >
> > > We thank the reviewer for their reply and their engagement. Please let us know if there is some specific weakness or way to improve the paper further!

---

### Official Review · Reviewer_E4hs · 2024-11-03

**Soundness:** 3
**Presentation:** 3
**Contribution:** 3
**Rating:** 6
**Confidence:** 2

**Summary:**

This paper provides a first connection between two federated learning approaches, one based on ADMM and another based on Variational Bayes (VB). Based on this connection, the paper proposes two variants that incorporate the full covariance information and functional regularization. The proposed methods improve the ADMM baseline in various benchmarks.

**Strengths:**

- The connection between ADMM and VB is novel and interesting, which opens up new avenues for improving the algorithm from a variational / Bayesian perspective.

- The overall empirical results are convincing, showing improvements of the proposed methods against the baseline.

**Weaknesses:**

The paper can be improved  by adding more discussions on complexity and limitations, as well as empirical results quantification. In particular, the key weaknesses are the following.


- missing discussion on limitations. What is the complexity of the extended VB approaches against the baselines, in particular when full covariance estimates are considered. Additionally, FedLap-Func is used in the setting where some inputs are shared globally. Can you discuss the practical implication of that?

- the empirical comparison needs more statistical rigorousness. Table 1 displays the average accuracy with standard deviation in parentheses, and top two performing algorithms are bolded. Can you elaborate how do you define the "top two"? A more standard practice in statistics is to highlight the top results within 2 *standard errors*, in that case I would imagine, for example, FedDyn and FedLap to be statistically equally performant on CIFAR10 with comm Round=50 (last row of the table 1).

**Questions:**

see the weakness section.

---

> ### Author Response · Authors · 2024-11-20
> **Author response**
>
> We thank the reviewer for their comments and review. We are glad that the reviewer appreciates the connection between ADMM and Bayes, and finds the empirical results overall to be convincing. We respond to the reviewer’s questions individually below. We have updated our paper on openreview, with changes highlighted in blue.
>
> **Q1: “What is the complexity of the extended VB approaches against the baselines, in particular when full covariance estimates are considered.”**
> We have added analysis as requested in the new submitted version, please see the [response to all reviewers](https://openreview.net/forum?id=ipQrjRsl11&noteId=Hcig5DGzFv) for the lines changed. To summarise, FedDyn and FedLap have the same computation and communication cost, while Fedlap-Cov and FedLap-Func increase the cost (but only slightly) due to the use of additional variables.
>
> **Q2: “FedLap-Func is used in the setting where some inputs are shared globally. Can you discuss the practical implication of that?”**
> Sharing inputs between local clients and the central server is common in Federated Distillation (FD) work (see eg Li and Wang (2019)). The focus of our work is to show theoretical links between the derived methods and FD algorithms generally. FD works normally randomly split an existing benchmark dataset (eg CIFAR-10) into a publicly-available part (that all clients and the server has access to) and a private part; in contrast, we assume that all parties have access to a random 1 input per class per client (which is significantly fewer datapoints than FD settings). FedLap-Func still performs well in our low-shared-data setting because, unlike FD algorithms, it also sends/uses weights. We discuss the limitations of this in lines 422-424 (new version lines 437-439): it introduces privacy issues, but as we use very few datapoints, it might be reasonable if we obtain prior permission from the owner(s) of the data. We hope that future work can build upon these insights further and show that FedLap-Func is competitive in full FD settings.
>
> **Q3: “the empirical comparison needs more statistical rigorousness.”**
> Currently, we simply bold the top two (mean) values, and do not bold if multiple (mean) values are the same (we have made this clearer in the new paper version’s Table 1 caption). We can alternatively bold methods that are within one standard deviation of the top-performing method. We note that previous papers usually do not run multiple random seeds at all (eg FedAvg, FedProx, FedDyn papers).

---

> > ### Comment · Reviewer_E4hs · 2024-11-26
> >
> > I would like to thank the authors for their reply. After consideration, I decided to keep my score.

---

> > > ### Author Response · Authors · 2024-11-27
> > > **Author response**
> > >
> > > We thank the reviewer for their reply and their engagement. Please let us know if there is some specific weakness or way to improve the paper further!

---

### Official Review · Reviewer_GZSC · 2024-11-03

**Soundness:** 2
**Presentation:** 2
**Contribution:** 2
**Rating:** 5
**Confidence:** 3

**Summary:**

The authors provide new insights connecting ADMM (Alternating Direction Method of Multipliers) and variational Bayesian (VB)-based approaches for federated learning. The main finding reveals that the dual variables utilized in ADMM naturally correspond to the 'site' parameters in VB.

**Strengths:**

1. connections between ADMM and VB-based approaches may be interesting for federated learning.

**Weaknesses:**

1. Whether the federated locals can be used to recover the global solution is not justified both theoretically and numerically.

2. Why do you want to connect ADMM to Bayesian? Bayesian learning is known to be the go-to method for conducting uncertainty quantification and non-convex optimization. The algorithm is only evaluated in the optimization perspective. The Bayes interpretation is only used as a preconditioner. If this is the only purpose, the connections to Bayes are not needed at all.

3. Bayes seems to be an important component in this paper. However, the connections to Bayes are not fully evaluated empirically. I recommend the authors to see how the uncertainty experiments are evaluated in [1,2,3].

[1]. Federated Learning via Posterior Averaging: A New Perspective and Practical Algorithms. ICLR. 2021.

[2]. On Convergence of Federated Averaging Langevin Dynamics. UAI 2024.

[3]. Bayesian Federated Learning with Hamiltonian Monte Carlo: Algorithm and Theory. JCGS. 2024

**Questions:**

Missing important literatures in Bayes federated learning:

[1]. Federated Learning via Posterior Averaging: A New Perspective and Practical Algorithms. ICLR. 2021.

[2]. On Convergence of Federated Averaging Langevin Dynamics. UAI 2024.

[3]. Bayesian Federated Learning with Hamiltonian Monte Carlo: Algorithm and Theory. JCGS. 2024

---

> ### Author Response · Authors · 2024-11-20
> **Author response**
>
> We thank the reviewer for their comments and review. We respond to the reviewer’s questions individually below. We have updated our paper on openreview, with changes highlighted in blue.
>
> **Q1: “Whether the federated locals can be used to recover the global solution is not justified both theoretically and numerically.”**
> We do have a theoretical justification in lines 201-204, 274-275 and 341-342 (new version lines 210-212, 291-292, 355-356) where we show that the stationary points of FedLap (and its variants) are the same as the global solution. In addition, in Appendix D2 we show empirically how FedLap correctly targets its global loss. We also show this numerically, for example, our experiments section (Section 4) shows how FedLap performs at least as well as FedDyn, and FedLap-Cov and FedLap-Func are even quicker to converge.
>
> **Q2: “Why do you want to connect ADMM to Bayesian? Bayesian learning is known to be the go-to method for conducting uncertainty quantification and non-convex optimization. The algorithm is only evaluated in the optimization perspective.” … “the connections to Bayes are not fully evaluated empirically. I recommend the authors to see how the uncertainty experiments are evaluated in [1,2,3].”**
> We disagree with the reviewer. Our focus on the paper is to draw methodological links between ADMM and Bayes (two historically-different fields), and show how this can be used to improve the optimisation itself. This is an important contribution. Also note that the extension of ADMM to preconditioned updates is not straightforward and often requires versions that use Bregman divergence. Our connection here shows that it can also be achieved by using Bayes where using covariances automatically leads to preconditioning by uncertainty estimates. This is a new result and has implications for learning with heterogenous data. For example, it suggests that the learning rates of clients should be modified according to their uncertainty.
>
> **Q3: “Missing important literatures in Bayes federated learning”.**
> We thank the reviewer for these references, and have added them to our paper (new version lines 113-115). All three papers [1,2,3] use MCMC algorithms in a federated learning setting, and so are orthogonal to the Variational Bayes approach we use in this paper.

---

> > ### Comment · Reviewer_GZSC · 2024-11-24
> >
> > Thank you for your response. I have slightly adjusted my rating. In a Bayesian setup, the extension to a preconditioned version is straightforward. However, I acknowledge that my knowledge of ADMM is limited, so I will leave the judgment of the novelty to other reviewers.

---

> > > ### Author Response · Authors · 2024-11-27
> > > **Author response**
> > >
> > > We thank the reviewer for their reply and their engagement. Please let us know if there is some specific weakness or way to improve the paper further!

---

### Official Review · Reviewer_8nbj · 2024-11-08

**Soundness:** 3
**Presentation:** 4
**Contribution:** 3
**Rating:** 8
**Confidence:** 3

**Summary:**

Federated learning is the problem of training a model where training data is distributed over clients and due to privacy must stay on clients. Given $K$ clients and a parameterzed model with paramters $w$, each client can optimize its parameters $w_k$ for $k=1,..,K$ and we need to merge the information to learn a global model $w_g$.

Many methods have been proposed and within which Alternative Direction Methods of Multiplier is a method that ensures the clients and the
global model converge to the same stationary point, the clients send a $v_k$ vector to the server that (loosly) shows the weight change from the global model $w_k - w_g$, the global model aggregates and updates and replies with the new global model $w_g$.

Similarly Bayesian methods learn an approximate posterior distribution over weights $q(w)$ where clients have their learn their own model $q_k()$ send the server an approximate likelihood function $\hat{t}(w)$. The server aggregates likelihoods, updates the global approximate posterior $q_g()$ and sends it back to the clients, Partitioned Variational Inference (PVI) is a recent work that constructs a common framework covering many prior work methods.

The paper describes the above frameworks. Then the authors show that for a particular choice of $q_k()$, $q_g()$ in the PVI framework, notably where they are Gaussian with a priori fixed variance so only the mean is learnt, corresponds almost exactly line by line with ADMM, i.e. ADMM is a MAP estimate of some specific Gaussian distributions. The authors propose FedLap-Cov that simply generalizes this observation by allowing learnable covariance matrices in the Gaussian distributions. Finally FedLap-func is proposed that effectively allows a subtle data sharing between client and server hence the server can perform weight optimization as well.

Experiments are performed across a range of standard benchmarks showing FedLap variants outperform both poor (FedAvg) and strong (FedDyn) baselines.

**Strengths:**

## Main Comments
- __intuitive demonstration of equivalence__: models trained by minimizing a loss function and a regularizer are often easily shown to correspond to a log likelihood and a log prior hence correspond to a MAP estimate. This work shows the same connection in a less obvious setting.
- __intuitive extension__: as corollary from above, generalizing the ADMM method for more general distributions makes perfect sense and allows a natural update equations, notably the precision matrix "weighted sum" of global and local models.
- __well written__: the paper takes the reader on a well paced journey through prior works ADMM, PVI using very readable mathematical notation (more readable than the original works in my view). FedLap is then presented to specifically bridge the gap between the prior discussion and the new discussion of FedLap-Cov and FedLap-Func.
- __extensive benchmarks__: UCI datasets, MNIST and Cifar10 experiments are provided and a deeper dive in to FedLap vs FedDyn is given in the appendix. Hyper-parameters were tuned for all methods.

## Minor comments
- I personally like that the authors avoid saying "our method" and opt for the more humble approach of using the method name "FedLap" throughout the paper.

**Weaknesses:**

I felt the paper was very good and only have minor comments.

### Minor Comments
- __limitations__ I may have missed this but limitations do not seem to be clearly discussed
- __computational complexity__ on P6 only discusses the diagonal covariance case which has linear added complexity. While if I understand correctly, the FedLap-Cov has a quadratic additional parameters, upper triangle of a cov matrix. A few concerns come to mind (1) one goal of federated learning is handling unstable client connections by minimising communication rounds, but presumably also minimising traffic, FedLap-Cov seems to add significant traffic (2) the scalability of the FedLap-Cov to many parameters is limited, e.g. resnet-50, although a quick look at past works suggest small applications like UCI datasets, Cifar10 and MNIST are common benchmarks (and not imagenet).
- __final model is point estimate__ I may have misunderstood, but I could not find explicit mention about what weights at inference time, I assume a single set of weights $w_g$ was used, there was no marginalising. I.e. FedLap methods learn a Bayesian distribution over weights, but the benchmarks have just used a point estimate.
- __Equation 13__: I apologize if I am mistaken but I was rather confused, the first $m_k$ may be a typo? As I understand, in a delta approximation, normally, we would make a 2nd order Taylor approximation of loss $l_k(w)$ around $w=m$. But to avoid headaches with a hessian term with both $m$ and $S$ interacting as we optimize, we freeze the hessian to a value from a prior iteration $w=m_{old}$ artificially removing the dependence on $m$, this is arguably not an accurate Taylor approximation anymore, is this a common practice?


### Possible Typos
- Equation after Eqn 9: $\mathbb{E}[\hat{t}_k(w)]$ missing $\log()$
- Equation 12 square brackets on the second term
- Equation 13: first $l_k(m_k)$ -> $l_k(m)$

**Questions:**

- for the benchmarks, the weights used in to make predictions were just the MAP estimate? Was there any marginalising predictions over the distributions of weights?
- Equation 13 uses gradients from a point in space that is not the centre of the Taylor approximation, is this principled or just a necessary hotfix?
- Equation 16: can the authors provide some context as to why the new weight space term is subtracted?

---

> ### Author Response · Authors · 2024-11-20
> **Author response**
>
> We thank the reviewer for their comments and review. We are glad that the reviewer found the paper well-written and intuitive to understand, and that there are extensive benchmarks in the paper. We respond to the reviewer’s questions individually below. We have updated our paper on openreview, with changes highlighted in blue.
>
> **Q1: “limitations do not seem to be clearly discussed”.**
>  A limitation is that the computation cost increases slightly when using more expressive posterior approximations, however this is accompanied by improvements, therefore may not be considered a direct limitation rather a feature of the Bayesian framework.
>
> **Q2: “computational complexity”.**
> We have added analysis as requested in the new submitted version, please see the [response to all reviewers](https://openreview.net/forum?id=ipQrjRsl11&noteId=Hcig5DGzFv) for the lines changed. To summarise, FedDyn and FedLap have the same computation and communication cost, while Fedlap-Cov and FedLap-Func increase the cost (but only slightly) due to the use of additional variables.
>
> **Q3: “final model is a point estimate” … “the weights used in to make predictions were just the MAP estimate?”**
> Yes, we use a point estimate as the final model in all experiments. We already see the benefit of adding a Bayesian viewpoint in the improved convergence using FedLap-Cov and FedLap-Func.
>
> **Q4: “Equation 13 … the first $m_k$ may be a typo” … “Equation 13 uses gradients from a point in space that is not the centre of the Taylor approximation, is this principled or just a necessary hotfix?”**
> Yes, the first $m_k$ is a typo, it should be $m$. Thank you for pointing this out (we have fixed it in the new version). The derivation in Eq 13 avoids higher-order derivatives during optimisation. For instance, another way to do this would be to use a Taylor series approximation around $q(w) = N(w; m, S^{-1})$ as usual, but assume that $H_k(m)$ does not depend on $m$. We have updated the text around Eq 13 to make this clear in the new version.
>
> **Q5: “Equation 16: can the authors provide some context as to why the new weight space term is subtracted?”**
> The intuition is that the contribution of the memory points is already added in the function-space term (first term in Eq 16), and so to avoid double-counting the contribution from the memory points, we need to subtract their contribution, which we do in weight-space (second-term in Eq 16). We have added a sentence after Eq 16 in the new version to add this intuition.
>
> **Typos.** We thank the reviewer for pointing these out! We have (i) fixed the missing log in the equation after Eq 9, (ii) changed Eq 12’s parentheses to be square brackets, and (iii) changed $m_k$ to $m$ in Eq 13.

---

> ### Comment · Reviewer_8nbj · 2024-11-22
> **Thank you for the Response**
>
> Thank you for the response and thank you for adding clear computational cost descriptions.
>
> In the revision, on line 295, it is stated that the cost of FedLapCov with covariance would be $2P + P^2/2$, I wander if it should it be $P + P^2/ 2$? Just a mean vector and half a cov matrix?
>
> After a read of the other reviews, I will keep my score as is.

---

> > ### Author Response · Authors · 2024-11-22
> > **Thanks**
> >
> > Thank you for your response. You are right that we need to store the mean and half of covariance (triangular matrix). This comes out to be $P + P(P+1)/2 = 3P/2 + P^2/2$. Thanks for correcting this.

---

### Official Review · Reviewer_QNup · 2024-11-08

**Soundness:** 3
**Presentation:** 3
**Contribution:** 3
**Rating:** 6
**Confidence:** 4

**Summary:**

The work provides a new link between two well-known federated learning approaches: i) Alternative Direction Methods of Multiplier (ADMM) and Partitioned Variational Inference (PVI). The driving idea is that PVI updates have a line-by-line correspondence to the ADMM algorithm, which is even more similar under the assumption of isotropic Gaussian distributions for the family $Q$ of variational densities. In this direction, the authors introduce FedLap, a Laplace-approximation based version of PVI. Such derivation suggests that there exists an almost identical duality between ADMM and the common Bayesian approach to the federated learning problem. Later, having established such a connection, two more variants of FedLap are introduced, the first one (FedLap-Cov) puts a bit more structure in the candidate set Q, such some improvement is observed given some increment in the computational cost. The second variant (FedLap-Func) extends ideas from (Khan & Swaroop, 2021) by allowing the FedLap update of the global server to be an optimization problem and assuming that some inputs are available between local and global servers. Empirical results confirm the hypotheses made and the improvement in the accuracy for the three new FedLap-based methods.

**Strengths:**

**[S1]** - The work is clear, well-written, and polished. I particularly liked the spirit of first building the connection between ADMM and PVI, for later proposing three new methods/variants of FedLap that incrementally improve the accuracy performance. For those familiar with the probabilistic perspective of federated learning, PVI, and Laplace approximation, the work and particularly the contributions are easy to follow and nice to be proven on the empirical results.

**[S2]** - The drawing of correspondence between ADMM and PVI is elegant, and for sure is a great novel contribution if it has not been done before (I think it hasn't). Despite the introduction of the diagonal covariance in FedLap-Cov made sense to me, and the adaptation of updates seemed also correct. I liked the final technique to ensure the positive definiteness of local and global precision matrices in the paragraphs after (14), quite interesting and useful.

**[S3]** - Despite I did not particularly see the advantages of FedLap-Func in Section 3.4, I see its utility and the spirit of even improving a bit more the performance of FedLap. Here, my experience with such methods is that the ratio improvement/computational-cost-increment is not as favorable as in the FedLap-Cov. I imagine that could have limited the experiments to 10-100 clients only.. I could be wrong on this aspect, though.

**Weaknesses:**

Despite the fact that I think the work has significant strengths in its current version for being accepted, I detect some unclear parts, or at least "corners" where there is not much light to understand what is going on. Some of these are:

**[W1]** - Computational cost is for sure one. While reading the paper, the missing aspect of the computational cost always rings the bell of an average reader. I say this mainly because the connection between ADMM and PVI is fantastic, but once the improvement is made with FedLap and its variants, it is not explicitly indicated what are we paying to get that extra improvement in the performance. In this direction, I want to say that I am just missing the analysis and the clarity in that regard of the problem.

**[W2]** - I'm fine with Eq. (15) and setting up the update as an optimization problem, but then I do not really know if sharing some soft labels or inputs between local clients and the central server is worth it.. It kind of modifies a bit the spirit of federated learning and introduces extra issues like privacy, etc, that are ignored imho.

**[W3]** - It makes a lot of sense to put the work in the context of Laplace approximation methods, BNNs, etc where basically the parameter times parameter dimension of matrices is a huge problem for scalability. Some light-light indication of this is said around L224-225, but overall, I feel that all of this between Eq. 10 and Eq. 14 is ignored, or at least not being highlighted for the reader (the dimensionality problem, or an indication of what are we considering here). I am particularly concerned about V_k and the client updates.

**[W4]** - Experiments are fine, but Table 1 is somehow limited. I basically missed some larger datasets, where there is not that much similarity or structure in the data as in MNIST or FMNIST (i.e. Imagenet or similar), and augmenting a bit the number of clients to see what is really the flexibility/resistance of new variants to such challenges.

**Questions:**

Some minor questions that I thought about while reading and writing the review:

**[Q1]** - I'm curious to know where the term ‘sites’ in L112 is coming from

**[Q2]** - How does this delta approximation work in L185? How does this approximation affect the general problem?

**[Q3]** - What is the dimension of the second auxiliary/dual variable? I'm always afraid of the use of preconditioning.. could the authors add a bit more details in that regard

**[Q4]** - I didn't understand very well what is indicated in L309

**[Q5]** - L342: M_k = null space, Isn’t this obvious?

**[Q6]** - For Algorithm in Eq. 17. How many training steps per round and updates?

---

> ### Author Response · Authors · 2024-11-20
> **Author response (1/2)**
>
> We thank the reviewer for their comments and review. We are glad that the reviewer found the paper well-written, and found the link between ADMM and PVI to be elegant and novel. We respond to the reviewer’s questions individually below. We have updated our paper on openreview, with changes highlighted in blue.
>
> **Q1: “Computational cost … I am just missing the analysis and the clarity.”**
> We have added analysis as requested in the new submitted version, please see the [response to all reviewers](https://openreview.net/forum?id=ipQrjRsl11&noteId=Hcig5DGzFv) for the lines changed. To summarise, FedDyn and FedLap have the same computation and communication cost, while Fedlap-Cov and FedLap-Func increase the cost (but only slightly) due to the use of additional variables.
>
> **Q2: “I do not really know if sharing some soft labels or inputs between local clients and the central server is worth it.. It kind of modifies a bit the spirit of federated learning and introduces extra issues like privacy, etc, that are ignored imho.”**
> Sharing inputs between local clients and the central server is common in Federated Distillation (FD) work (see eg Li and Wang (2019)). The focus of our work is to show theoretical links between the derived methods and FD algorithms generally. FD works normally randomly split an existing benchmark dataset (eg CIFAR-10) into a publicly-available part (that all clients and the server has access to) and a private part; in contrast, we assume that all parties have access to a random 1 input per class per client (which is significantly fewer datapoints than FD settings). FedLap-Func still performs well in our low-shared-data setting because, unlike FD algorithms, it also sends/uses weights. We discuss the limitations of this in lines 422-424 (new version lines 437-439): as you say, it introduces privacy issues, but as we use very few datapoints, it might be reasonable if we obtain prior permission from the owner(s) of the data. We hope that future work can build upon these insights further and show that FedLap-Func is competitive in more regular FD settings.
>
> **Q3: “scalability … I feel that all of this between Eq. 10 and Eq. 14 is ignored, or at least not being highlighted for the reader (the dimensionality problem, or an indication of what are we considering here).” … “What is the dimension of the second auxiliary/dual variable? I'm always afraid of the use of preconditioning.. could the authors add a bit more details in that regard”.**
> In general, the second dual variable $V_k$ has dimension PxP, where P is the number of parameters of the model. However, as indicated in line 224 (new version line 238), we only consider diagonal approximations in our experiments, which we already show performs very well in FedLap-Cov (and as is common for both Laplace and variational neural networks), although other approximations are possible (eg block-diagonal K-FAC). In the diagonal case, $V_k$ has dimension P (same as the dimension of $w$ and of the first dual variable $v_k$). As requested, we have added/highlighted this point in lines 273-274 of the new paper version.
>
> **Q4: “Experiments are fine, but Table 1 is somehow limited. I basically missed some larger datasets, where there is not that much similarity or structure in the data as in MNIST or FMNIST … and augmenting a bit the number of clients.”**
> We understand the point. We use similar experiments to other works in this area (especially cross-silo works as opposed to cross-device). Other reviewers (8nbj, E4hs, kAKo) also found the experiments to be sufficient, but we agree that larger experiments may support the paper's argument even more.
>
> **Q5: “I'm curious to know where the term ‘sites’ in L112 is coming from”**
> ‘Sites’ are used in Bayesian message passing works to describe ‘locations’ where there is relevant information, and messages are passed to/from these sites. For example, this is seen in Belief Propagation and Expectation Propagation (Minka, 2001), where the $t_k$ are referred to as the ‘sites’.
>
> **Q6: “How does this delta approximation work in L185? How does this approximation affect the general problem?”**
> As described in line 193 (new version line 201), this approximates an expectation of a function by the value at the mean: $\mathbb{E}_q[g(w)] \approx g(m)$. In our setting, this approximation allows us to recover (a local) Laplace approximation as a special case of (the more global) Variational-Bayes; see Opper and Archambeau (2009) for the local-global connection.

---

> ### Author Response · Authors · 2024-11-20
> **Author response (2/2)**
>
> **Q7: “I didn't understand very well what is indicated in L309.”**
> We add and subtract two terms that are equal at the point $w_k$: the true loss over inputs in $M_k$, and the Gaussian contribution over those points (as derived in Khan et al. (2019)).
> The detailed derivation is in Appendix C1: we show in Eq 23 how terms cancel so that we do not need access to true labels, but only to soft labels. In the new version of the paper, we removed some of the detail in Sec 3.4 to avoid confusion (the details explaining the derivation require more explanation, as is in App C1). Please let us know if this did not make sense!
>
> **Q8: “L342: M_k = null space, Isn’t this obvious?”**
> Yes, when $M_k$ is the empty set, then FedLap-Func recovers FedLap. Existing Federated Distillation works do not have this property: they are stand-alone algorithms. Please let us know if you meant something different here.
>
> **Q9: “For Algorithm in Eq. 17. How many training steps per round and updates?”**
> We have now added how many training steps per round (and learning rate) for the global server optimisation in App E, thank you for pointing this out. In summary, we use 5000 updates at the server (500 for CIFAR), but did not hyperparameter sweep over this value: we stress that this will usually not be a problem because in most federated learning settings, the global server has significantly more compute resources than the local clients; also, the number of input points at the server much smaller than the data at local clients (as discussed in the computation complexity section, new version lines 364-372).

---

### Author Response · Authors · 2024-11-20
**Overview of response**

We thank all the reviewers for their reviews, and for their time. We appreciate the comments and believe the paper will be stronger as a result. In particular, we are pleased reviewers found the theory (linking federated ADMM to Bayes) to be novel (QNup, E4hs, kAKo) and interesting/significant (QNup, GZSC, E4hs, kAKo), the paper to be well-written (QNup, 8nbj), and the experiments to be convincing/extensive (8nbj, E4hs, kAKo). We stress that, to the best of our knowledge, we are the first work to draw links between ADMM and Bayes, and hope that future work will build on this.

We have uploaded an updated draft with suggested changes by reviewers (changes are highlighted in blue in the uploaded pdf).

Reviewers asked us to make the computational complexity of new algorithms explicit.
We have added/changed the following lines in the paper:
1. FedDyn and FedLap have the same computation and communication cost. A paragraph has been added in the new version’s lines 225-229 to show this precisely.
2. FedLap-Cov and FedLap-Func increase computational cost and communication cost (but only slightly) due to the additional variables. We have added precise notation to our previous text in lines 294-299 for FedLap-Cov and lines 364-372 for FedLap-Func.

We respond to other reviewer comments individually. We look forward to responding to any other concerns or comments that the reviewers have during the discussion period!

---

### Meta-Review · Area_Chair_T4je · 2024-12-20

**Metareview:**

This paper provides a novel connection between ADMM and variational inference (PVI) in the context of federated learning and develops new algorithms based on this relationship. By leveraging the similarity between ADMM and PVI under the assumption of Gaussian distributions, the authors introduce new algorithms, FedLap, which are based on Laplace approximations. The effectiveness of these algorithms is demonstrated on benchmark datasets. As for the weakness, the method suffers from the increased computational costs compared to existing methods. Despite this limitation, all reviewers positively evaluated the connection established in this work and the utility of the proposed algorithms. Therefore, I recommend acceptance.

**Additional Comments On Reviewer Discussion:**

Reviewer 8nbj and Reviewer QNup raised concerns regarding the increased computational costs, but these were resolved through additional discussions provided by the authors. Similarly, Reviewer E4hs raised questions about the relationship with Bayesian methods, which were also addressed through the authors’ explanations. Given these resolutions, I believe there are no issues preventing the paper from being published.

---

### Decision · Program_Chairs · 2025-01-22

Accept (Poster)